# Deep Learning for Time Series Forecasting: Advances and Open Problems

**Angelo Casolaro** †, **Vincenzo Capone** †, **Gennaro Iannuzzo** † and **Francesco Camastra** *,†

Department of Science and Technology, Parthenope University of Naples, Centro Direzionale Isola C4, 80143 Naples, Italy; angelo.casolaro001@studenti.uniparthenope.it (A.C.); vincenzo.capone002@studenti.uniparthenope.it (V.C.); gennaro.iannuzzo001@studenti.uniparthenope.it (G.I.)
* Correspondence: francesco.camastra@uniparthenope.it
† These authors contributed equally to this work.

**Abstract:** A time series is a sequence of time-ordered data, and it is generally used to describe how a phenomenon evolves over time. Time series forecasting, estimating future values of time series, allows the implementation of decision-making strategies. Deep learning, the currently leading field of machine learning, applied to time series forecasting can cope with complex and high-dimensional time series that cannot be usually handled by other machine learning techniques. The aim of the work is to provide a review of state-of-the-art deep learning architectures for time series forecasting, underline recent advances and open problems, and also pay attention to benchmark data sets. Moreover, the work presents a clear distinction between deep learning architectures that are suitable for short-term and long-term forecasting. With respect to existing literature, the major advantage of the work consists in describing the most recent architectures for time series forecasting, such as Graph Neural Networks, Deep Gaussian Processes, Generative Adversarial Networks, Diffusion Models, and Transformers.

**Keywords:** short-term forecasting; long-term forecasting; recurrent neural networks; temporal convolutional neural networks; graph neural networks; deep gaussian processes; transformers; time series benchmarking; generative adversarial networks; diffusion models

## 1. Introduction

A time series is a sequence of data enumerated in time order. Time series are used to study how certain measures, e.g., air pollution data [1], ozone concentration [2], plant growth [3], sunspots [4], Dow Jones and Nasdaq indices [5], and electricity consumption [6], evolve over time. Time series forecasting estimates values that a time series takes in the future, allowing the implementation of decision-making strategies, e.g., abandonment of fossil fuels to reduce the surface temperature of the Earth. Specifically, time series forecasting is very relevant for the *energy domain* (e.g., electricity load demand [7,8], solar and wind power estimation [9,10]), *meteorology* (e.g., prediction of wind speed [11], temperature [12,13], humidity [12], precipitation [13,14]), *air pollution monitoring* (e.g., prediction of $PM_{2.5}$, $PM_{10}$, $NO_2$, $O_3$, $SO_2$, and $CO_2$ concentrations [12,15,16]), the *finance domain* (e.g., stock market index and shares prediction [17,18], the stock price [19,20], exchange rate [21,22]), *health* (e.g., prediction of infective diseases diffusion [23], diabetes mellitus [24], blood glucose concentration [25], and cancer growth [26]), *traffic* (e.g., traffic speed and flow prediction [27–30]), and *industrial production* (e.g., petroleum production [31], remaining life prediction [23,32,33], industrial processes [34], fuel cells durability [35], engine faults [36]). Deep learning algorithms are currently the leading methods in machine learning due to their successful application to many computer science domains (e.g., *computer vision*, *natural language processing*, *speech recognition*). In recent years, there has been a growth of interest in the application of deep learning to time series forecasting, due to deep learning's capability to grasp the nonlinear relations among input data (i.e., time series samples). To the best

of our knowledge, there are several reviews on deep learning for time series forecasting (e.g., [37–43]), but they have the following major shortcomings: they lack a description of *Transformer* and its variants; they do not provide a clear distinction between models for short-term and long-term forecasting, there are no sections on short-term and long-term forecasting benchmarks; they do not cover the most recent deep learning strategies for short-term forecasting (e.g., *Graph Neural Networks*, *Deep Gaussian Processes*, *Generative Adversarial Networks*, and *Diffusion Models*). The aim of this work is to provide a comprehensive survey of recent deep learning approaches for time series forecasting, underlining the advances and the open problems for each reviewed deep learning architecture. Specifically, the survey focuses on works about machine learning applied to time series forecasting that are not older than 2016, for the sake of length. The paper is organised in the following sections: in Section 2, the foundations of deterministic time series are introduced; Section 3 describes deep learning architectures for short-term forecasting, i.e., *Convolutional Neural Networks*, *Temporal Convolutional Networks*, *Recurrent Neural Networks* (RNNs), *Graph Neural Networks*, *Deep Gaussian Processes*, *Generative Adversarial Networks*, and *Diffusion Models*; Section 4 discusses long-term forecasting architectures, i.e., the *Transformer* architecture and its time series-based variants; Section 5 reviews other heterogeneous architectures for both short-term and long-term forecasting; benchmarking for short-term and long-term time series forecasting is presented in Section 6; in Section 7, some conclusions are drawn and future possible developments are discussed; finally, in the appendix are reported the main mathematical notation used in the work and a description of the main diffusion model foundations.

## 2. Deterministic Time Series

A time series is called a *univariate time series* if all its samples are scalar; otherwise, if all samples are vectors, it is called a *multivariate time series*. A time series is defined as *stationary* when the dynamical process that generated it does not change over time, otherwise, it is *non-stationary*. A *deterministic stationary time series* $x_t$, with $t = 1, \ldots, L$, can be described by an autoregressive model as follows:

$$x_{t+p} = f(x_{t-1}, \ldots, x_{t-q}) + \epsilon_{t+p} \quad \forall p \in [0, P] \tag{1}$$

where $f(\cdot)$ and $q$ are called *skeleton* and *model order* of time series, i.e., how many past samples are required to model the time series adequately, respectively, and $\epsilon_{t+p}$ represents an indeterminable part originated either from unmodeled dynamics of the process or from real noise. $P$ defines the *prediction length*, i.e., how many future samples should be predicted. Figure 1 gives a graphical representation of an autoregressive model.

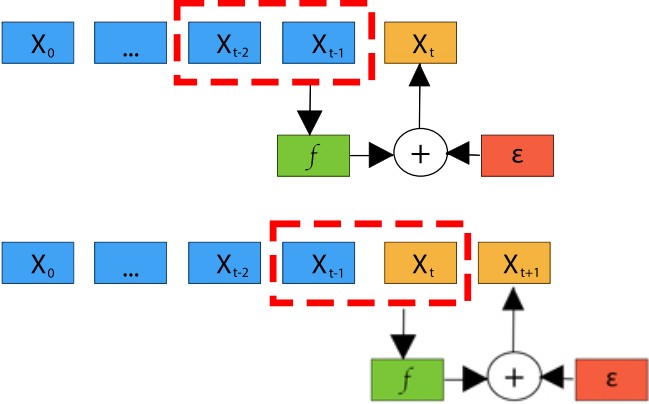

**Figure 1.** An example of an autoregressive model for forecasting based on deterministic stationary time series. In the figure, the model order is $q = 2$ and the prediction length is $P = 0$ (i.e., it is a one-step ahead forecasting problem).

If $P = 0$, the autoregressive model defines the so-called *one-step ahead forecasting*, otherwise, a prediction length $P > 0$ specifies a *multi-step ahead forecasting* problem. Moreover, multi-step ahead forecasting can be further divided into *short-term* and *long-term* forecasting. In the literature, the typical threshold value of prediction length $P$ between short-term and long-term forecasting ranges between 2 and 48 [44]. Finally, for the sake of clarity, in this work, one-step ahead forecasting is included in short-term forecasting.

## 3. Deep Learning Models for Short-Term Forecasting

In short-term forecasting, the skeleton of time series can be approximated by the following deep learning models, which are described below. The section is organised as follows. Firstly, *Convolutional Neural Networks* (Section 3.1), and *Temporal Convolutional Networks* (Section 3.1.2) are introduced. Furthermore, *Recurrent Neural Networks* (Section 3.2) are described, including *Elman RNNs* (Section 3.2.1), *Echo State Networks* (Section 3.2.3), *Long Short-Term Memory* (Section 3.2.4), and *Gated Recurrent Units* (Section 3.2.5). Successively, hybrids and variants (Section 3.3) of the aforementioned architectures are briefly reviewed. Moreover, *Graph Neural Networks* (Section 3.4), *Deep Gaussian Processes* (Section 3.5), and generative models (Section 3.6), i.e., *Generative Adversarial Networks* (Section 3.6.1) and *Diffusion Models* (Section 3.6.3), are discussed. Finally, for each reviewed model, drawbacks and limitations are discussed in proper sections.

### 3.1. Convolutional Neural Networks

*Convolutional Neural Networks* (CNNs) [45], as shown in Figure 2, have a deep architecture generally formed by *convolution*, *pooling*, and *fully connected* layers. CNNs have three main peculiarities: *local connectivity*, *shared weights* and *translation equivariance*. Local connectivity resides in each CNN neuron being connected only to its exclusive input region, the so-called *receptive field*. Besides, the neurons of a given layer share the same weight matrix. Translation equivariance is the ability of CNNs to detect certain patterns, regardless of the position they have in the input image. *1D convolution* (see Figure 3) of an input sequence $\mathcal{X} = [x_1, \ldots, x_L]$ with a given *kernel w* with *size q* is defined as:

$$y(t) = (w * \mathcal{X})(t) = \sum_{a=-q/2}^{q/2} w(a)\mathcal{X}(t-a) \quad \forall t \in [1, \ldots, L] \tag{2}$$

It is worthwhile to remark that in the autoregressive approach, the kernel size $q$ corresponds to the model order, generally fixed using *model selection techniques* (e.g., *cross-validation*) [46]. Moreover, CNN can stack different convolutional layers in order to transform the input data (i.e., the past time series values) into a more suitable higher-level representation for the forecasting task. CNN time series forecasting applications are described in Table 1.

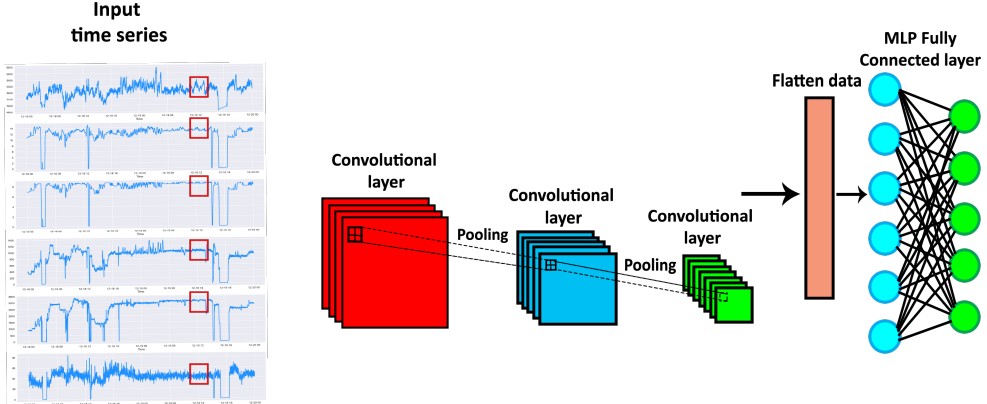

**Figure 2.** An example of Convolutional Neural Network applied to time series forecasting. The red, the blue and the green boxes represent CNN's convolutional layers.

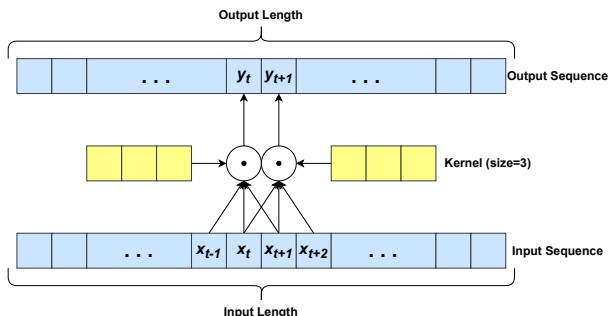

**Figure 3.** Example of 1D convolution using a kernel with size $k = 3$. The scalar product is denoted by ●. The yellow boxes denote the *learned kernel*.

**Table 1.** Recent applications on time series forecasting using Convolutional Neural Networks.

| Ref. | Year | Application |
|:---:|:---:|:---:|
| [47] | 2017 | ETFs prices |
| [48] | 2018 | Electricity consumption |
| [10] | 2018 | Solar power and electricity load |
| [6] | 2018 | Electricity consumption |
| [7] | 2018 | Electricity price |
| [49] | 2019 | Electricity price and load forecasting |
| [50] | 2019 | Building-level load |
| [12] | 2023 | $CO_2$/Temperature/Humidity |

### 3.1.1. Shortcomings of Convolutional Neural Networks

The major drawback of CNNs for time series forecasting is the use of *Euclidean kernels* [51], since only a continuous and commonly short time series subsequence is considered at a time by an Euclidean kernel. However, in time series forecasting tasks it may be useful to extract representative features by analysing *non-contiguous* and *farther* time series samples in the past. This limitation is overcome by *Temporal Convolutional Networks* (see Section 3.1.2), with the use of *causal* and *dilated* convolutions, and by *Graph Neural Networks* (see Section 3.4), with a properly designed adjacency matrix.

### 3.1.2. Temporal Convolutional Networks

*Temporal Convolutional Networks* (TCNs) were proposed for *action segmentation and detection* by Lea et al. [52]. In a nutshell, a TCN is composed of cascaded 1D convolutional layers that allow mapping arbitrarily long inputs onto output sequences of the same length. In contrast to traditional CNNs, TCNs perform *causal* and *dilated* convolutions. Unlike *1D convolution* (see Equation (2)), in *causal 1D convolution* (see Figure 4) the output element at time $t$ is yielded by the convolution between the kernel and *past input elements* only, namely $[x_{t-1}, x_{t-2}, \ldots, x_{t-q}]$, where $q$ is the kernel size that corresponds to the model order in an autoregressive approach (see Section 3.1). In detail, causal 1D convolution is defined as follows:

$$y(t) = (w * \mathcal{X})(t) = \sum_{a=1}^{q} w(a)\mathcal{X}(t - a) \quad \forall t \in [1, \ldots, L] \tag{3}$$

A *dilated 1D convolution* (see Figure 5) differs from a 1D convolution due to the introduction of a *dilation factor d*. The dilation, applied to convolution, is equivalent to considering a fixed step between two adjacent kernel elements. In particular, dilated causal 1D convolution can be defined as:

$$y(t) = (w * \mathcal{X})(t) = \sum_{a=1}^{q} w(a)\mathcal{X}(t - d \cdot a) \quad \forall t \in [1, \ldots, L] \tag{4}$$

When $d = 1$, a dilated 1D convolution is reduced to a 1D convolution. Dilated convolutions allow the networks to have a large receptive field, i.e., to capture information into the far past, by a logarithmic growth of the number $g$ of convolutional layers, as stated in:

$$g = \left\lceil \log_b \left( \frac{(n-1)(b-1)}{(q-1)} + 1 \right) \right\rceil \tag{5}$$

where $b$ is the *logarithmic base* of the dilation factor $d_i$ for the $i$-th convolutional layer (namely, $d_i = b^i$). TCN time series forecasting applications are reported in Table 2.

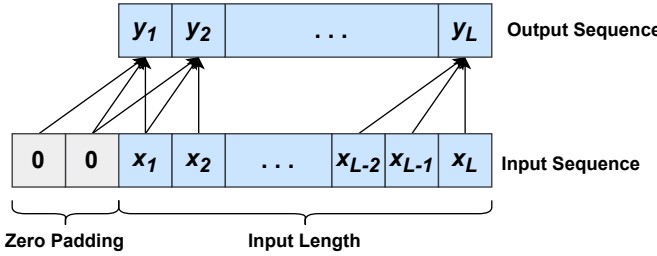

**Figure 4.** Causal 1D convolution with a kernel of size $q = 3$. Zero padding elements are added at the beginning of the input sequence in order to have an output sequence with the same length as the input sequence.

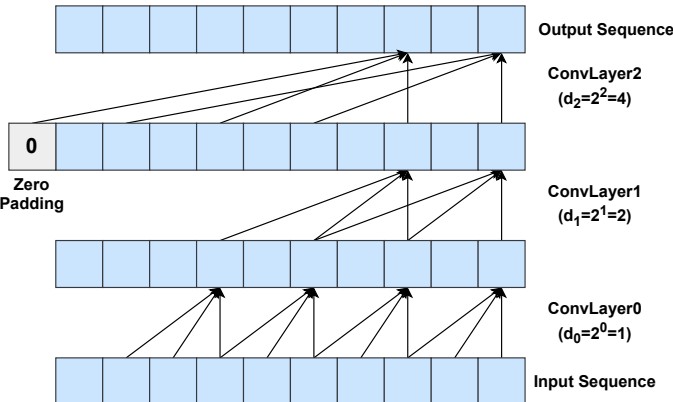

**Figure 5.** Example of a dilated causal 1D convolution with three layers using a dilation base $b = 2$ and a kernel size $q = 3$. Zero padding is used to preserve the input sequence length.

**Table 2.** Time series forecasting applications using Temporal Convolutional Networks.

| Ref. | Year | Application |
|------|------|-------------|
| [53] | 2018 | Stock market |
| [15] | 2019 | Beijing $PM_{2.5}$ |
| [30] | 2019 | Traffic |
| [54] | 2020 | National electric demand and power demand |
| [9]  | 2020 | Wind power generation |
| [55] | 2020 | Weather |
| [11] | 2022 | Wind speed |

TCNs share with CNNs most shortcomings (see Section 3.1.1), with the only exception of the Euclidean kernel.

### *3.2. Recurrent Neural Networks*

*Recurrent Neural Networks* (RNNs) [45] aim to explore the relations between the current time series samples and the past ones. An RNN processes one time series sample at a time to approximate the skeleton $f(\cdot)$ and determine the model order $q$ implicitly. While in Equation (1) the model order $q$ is assumed to be constant, RNNs exhibit a dynamic temporal behaviour, as they do not require a fixed model order $q$ and they auto-determine how far back to look in the past. An RNN is composed of recurrent layers, which process one input sample at a time. The earliest RNN applications for time series forecasting [56,57] are replaced by more specific and sophisticated recurrent architectures, that is, *Elman Recurrent Neural Networks* [58], *Echo State Networks* [59], *Long Short-Term Memory Networks* [60], and *Gated Recurrent Units* [61].

#### 3.2.1. Elman Recurrent Neural Networks

*Elman* [58], *Williams-Zipser* [62], and *Jordan* [63] RNNs are the earliest Recurrent Neural Networks properly designed to handle temporal patterns in time series. In particular, *Elman RNN* (ERNN) uses a recurrent layer, which yields an output $\vec{h}_t$ that depends on the current sample $\vec{x}_t$ and the previous output $\vec{h}_{t-1}$ by a function $g(\cdot)$ and a generic set of time-shared parameters $\omega$, as described:

$$\vec{h}_t = g_\omega(\vec{x}_t, \vec{h}_{t-1}) \tag{6}$$

where $\vec{h}_{t-1}$ is produced by the same recurrent layer, i.e.,:

$$\vec{h}_{t-1} = g_\omega(\vec{x}_{t-1}, \vec{h}_{t-2}) \tag{7}$$

and so on. The basic recurrent layer, often called a *vanilla cell*, works like a fully connected layer with a fixed number of units, jointly applied to the current input $\vec{x}_t$ and the last output $\vec{h}_{t-1}$:

$$\vec{h}_t = g\left(V\vec{x}_t + W\vec{h}_{t-1} + \vec{b}\right) \tag{8}$$

In this case, the set of parameters of a recurrent layer is $\omega = \{V, W, \vec{b}\}$, where $V$ is the input-recurrent weight matrix, $W$ is the recurrent-recurrent weight matrix, and $\vec{b}$ is the bias vector. In Equation (8), $g(\cdot)$ is a nonlinear activation function, usually a hyperbolic tangent. ERNN time series forecasting applications are summarised in Table 3.

**Table 3.** Elman RNN applications for time series forecasting.

| Ref. | Year | Application |
|------|------|-------------|
| [64] | 2017 | Electricity load |
| [65] | 2018 | Electricity load |
| [66] | 2018 | Energy consumption |
| [14] | 2019 | Monthly precipitation |
| [16] | 2021 | Air Quality Index |

#### 3.2.2. Shortcomings of Recurrent Neural Networks

Recurrent neural networks based on the vanilla cell suffer from unstable training, which prevents the network from grasping *long-term dependencies*. Recurrent networks, like most neural networks, are trained by *gradient descent* [67], and *backpropagation* [67] (Backpropagation is denoted *Backpropagation Through Time* (BPTT), when applied to recurrent neural networks) is used to compute the gradient of the loss function w.r.t. the network's weights. When back-propagation is applied to deep networks, the problems of *vanishing* or *exploding gradients* [45] may arise. As the error gradient is back-propagated, some of its components might either get very small, giving a negligible contribution to the corresponding weight update, or very large, leading, in this way, to unstable training. Over the years, several approaches have been proposed to cope with unstable gradients.

Among the most successful approaches are *reservoir computing* methods, e.g., *Echo State Networks* [59] (see Section 3.2.3), and *gated cells*, e.g., *Long Short-Term Memory* (LSTM) cells [60] and *Gated Recurrent Units* (GRU) [61]. A gated cell controls how much input information flows through the layer and prevents derivatives from vanishing or getting large.

### 3.2.3. Echo State Networks

*Echo State Networks* (ESNs) were suggested by *H. Jaeger* [59] in 2001 as a variant of *ERNNs*. ESNs are really effective in dealing with *chaotic multivariate time series* [68]. In addition, these networks mitigate the unstable gradient problem and are more computationally efficient due to the use of fixed, random weight matrices for the recurrent layers. Based on the vanilla cell of ERNNs (see Equation (8)), ESNs avoid backpropagation on the recurrent layer by setting *V* and *W* as fixed (i.e., non-trainable) random matrices. Furthermore, a given *sparsity level* is considered in matrix *W*. Although random matrices are an advantage of ENSs to mitigate the unstable gradient problem, they, at the same time, represent a major ESNs shortcoming since they make particularly difficult the application of common interpretability approaches, e.g., *gradient-based approaches* [69,70]. ESN time series forecasting applications are described in Table 4.

**Table 4.** ESN applications on time series forecasting.

| Ref. | Year | Application |
|:---:|:---:|:---:|
| [71] | 2017 | Fuel cell voltage ageing |
| [32] | 2017 | Health of automotive batteries |
| [72] | 2017 | Slugging flow phenomenon |
| [13] | 2017 | Temperature/Rainfall |
| [73] | 2018 | Lorenz/Rossler/Sunspot-Runoff |
| [34] | 2019 | Industrial processes |
| [35] | 2019 | Fuel cell durability |
| [74] | 2019 | Photovoltaic voltage |
| [75] | 2020 | Electricity load |
| [76] | 2020 | Electricity load |
| [77] | 2020 | Energy consumption/Wind power generation |
| [78] | 2020 | Temperature of exhaust gas |
| [36] | 2020 | Faults in airplane engines |
| [79] | 2020 | Multiple time series |
| [25] | 2020 | Blood glucose concentration |
| [80] | 2021 | Multiple time series |
| [81] | 2021 | Electrical load |
| [16] | 2021 | Air Quality Index |
| [82] | 2022 | Chaotic time series |

### 3.2.4. Long Short-Term Memory

*Long Short-Term Memory* (LSTM) Networks [60] were originally proposed for *natural language modelling*. The LSTM cell (see Figure 6) uses three gating mechanisms: *input*, *forget* and *output gates*. Firstly, the input of LSTM layers, which is composed of the current input $\vec{x}_t$ and the output $\vec{h}_{t-1}$ from the last time step are is combined into the *current input* vector $\vec{i}_t$ as follows:

$$\vec{i}_t = \gamma(W_i \vec{x}_t + U_i \vec{h}_{t-1} + \vec{b}_i) \tag{9}$$

where $\gamma(\cdot)$ can be any *sigmoidal function* (e.g., *logistic* or *hyperbolic tangent* ones) and $\{W_i, U_i, \vec{b}_i\}$ is a set of parameters. Furthermore, the three gates are computed as:

$$\vec{g}_t = \sigma(W_g \vec{x}_t + U_g \vec{h}_{t-1} + \vec{b}_g) \tag{10}$$

$$\vec{f}_t = \sigma(W_f \vec{x}_t + U_f \vec{h}_{t-1} + \vec{b}_f) \tag{11}$$

$$\vec{o}_t = \sigma(W_o \vec{x}_t + U_o \vec{h}_{t-1} + \vec{b}_o) \tag{12}$$

where $\sigma(\cdot)$ is the logistic function, $\vec{g}_t$, $\vec{f}_t$, $\vec{o}_t$ are the input, forget, and output gates, respectively, and $\{W_g, U_g, \vec{b}_g\}$, $\{W_f, U_f, \vec{b}_f\}$, $\{W_o, U_o, \vec{b}_o\}$ are the respective sets of parameters. Furthermore, the LSTM's *inner state* $\vec{c}_t$ is updated by a linear combination of $\vec{i}_t$ and the *previous inner state* $\vec{c}_{t-1}$:

$$\vec{c}_t = \vec{g}_t \odot \vec{i}_t + \vec{f}_t \odot \vec{c}_{t-1} \tag{13}$$

where $\odot$ is the *element-wise product*. Finally, the output $\vec{h}_t$ of a LSTM layer is defined as:

$$\vec{h}_t = \vec{o}_t \odot \tanh(\vec{c}_t) \tag{14}$$

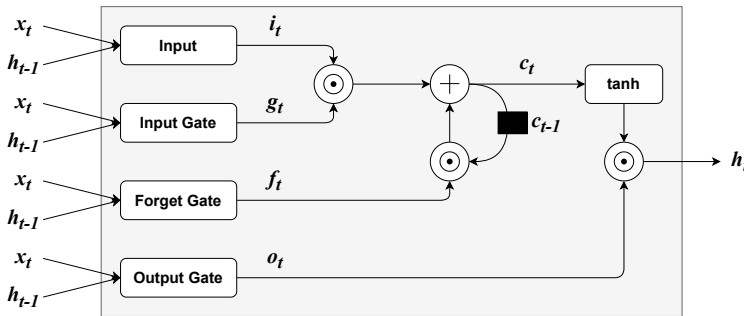

**Figure 6.** Long-Short Term Memory cell architecture.

LSTM time series forecasting applications are described in Table 5.

**Table 5.** LSTM applications on time series forecasting.

| Ref. | Year | Application |
|---|---|---|
| [17] | 2016 | Stock market |
| [83] | 2016 | Electricity load |
| [84] | 2016 | Traffic flow |
| [19] | 2017 | Stock prices |
| [85,86] | 2017 | Stock market |
| [87] | 2017 | Electricity load |
| [88] | 2017 | Air quality |
| [26] | 2018 | Forecasting Cancer Growth |
| [89,90] | 2018 | Stock market |
| [20] | 2018 | Stock prices |
| [7] | 2018 | Electricity price |
| [24] | 2018 | Diabetes mellitus |
| [91] | 2018 | Rainfall-runoff modelling |
| [92] | 2018 | Predicting water table depth |
| [93,94] | 2018 | Electricity load |
| [33] | 2018 | Life prediction of batteries |
| [10] | 2018 | Solar power and electricity load |
| [95] | 2018 | Solar intensity |
| [96] | 2018 | Air quality |
| [97] | 2019 | UCI data sets |
| [98] | 2019 | Building load |
| [31] | 2019 | Petroleum production |
| [14] | 2019 | Monthly precipitation |
| [99] | 2019 | Weather forecasting |

**Table 5.** *Cont.*

| Ref. | Year | Application |
|------|------|-------------|
| [18] | 2020 | Stock market |
| [100] | 2020 | COVID-19 |
| [79] | 2020 | Multiple time series |
| [101] | 2021 | Weather/Air Quality/Clinical data |
| [16] | 2021 | Air Quality Index |
| [102] | 2022 | Financial markets |
| [12] | 2023 | $CO_2$/Temperature/Humidity |

### 3.2.5. Gated Recurrent Units

RNNs based on *Gated Recurrent Units* (GRUs) [61] were introduced for *Statistical Machine Translation*. A GRU layer, as shown in Figure 7, uses two gating mechanisms: an *update* and a *reset gate*. Both the reset and the update gate depend on $\vec{x}_t$ and $\vec{h}_{t-1}$. Analogously to LSTM, the reset gate $\vec{r}_t$ and the update gate $\vec{u}_t$ are computed as:

$$\vec{r}_t = \sigma\left(W_r\vec{x}_t + U_r\vec{h}_{t-1} + \vec{b}_r\right) \tag{15}$$

$$\vec{u}_t = \sigma\left(W_u\vec{x}_t + U_u\vec{h}_{t-1} + \vec{b}_u\right) \tag{16}$$

where $\sigma(\cdot)$ is the logistic function and the rest of the parameters have the same meaning as the LSTM (see Section 3.2.4). Furthermore, an intermediate output $\vec{h}'_t$ is given by:

$$\vec{h}'_t = \tanh\left(W\vec{x}_t + U(\vec{r}_t \odot \vec{h}_{t-1}) + \vec{b}\right) \tag{17}$$

where $\{W, V, \vec{b}\}$ is an additional set of parameters and $\odot$ is the element-wise product. Finally, the output $\vec{h}_t$ of the GRU layer is given by the sum of $\vec{h}'_t$ and $\vec{h}_{t-1}$, weighted element-wise by the update gate:

$$\vec{h}_t = \vec{u}_t \odot \vec{h}_{t-1} + (\vec{e} - \vec{u}_t) \odot \vec{h}'_t \tag{18}$$

where $\vec{e}$ is a column vector whose elements are all equal to 1. GRU time series forecasting applications are described in Table 6.

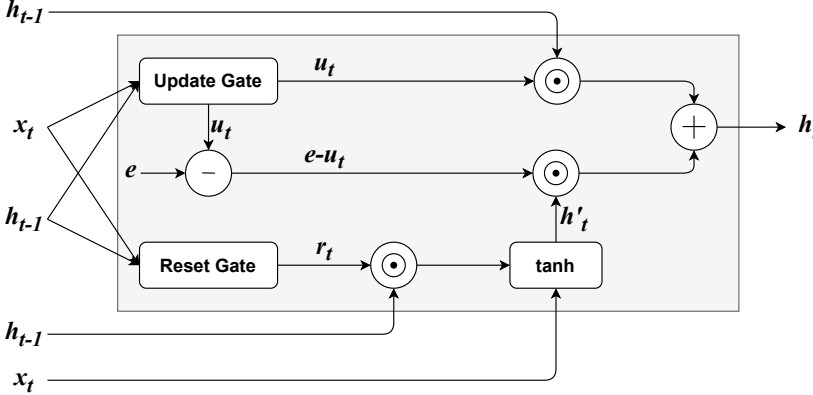

**Figure 7.** Architecture of a GRU cell. The column vector $\vec{e}$ is composed of elements that are all equal to 1.

**Table 6.** Applications on time series forecasting using GRU-based Recurrent Neural Networks.

| Ref. | Year | Application |
|------|------|-------------|
| [84] | 2016 | Traffic flow |
| [8] | 2017 | Electricity load |
| [103] | 2018 | Photovoltaic forecasting |
| [7] | 2018 | Electricity price |
| [24] | 2018 | Diabetes mellitus |
| [97] | 2019 | UCI data sets |
| [79] | 2020 | Multiple time series |
| [104] | 2021 | Air quality/Stock prices/Household electric power |

### 3.2.6. Shortcomings of LSTMs and GRUs

It has to be remarked that, even if training is stable, it can be hard for recurrent networks to learn dependencies between distant sequence samples. For instance, a recurrent network that generates an output sequence starting from an input sequence is shown in Figure 8.

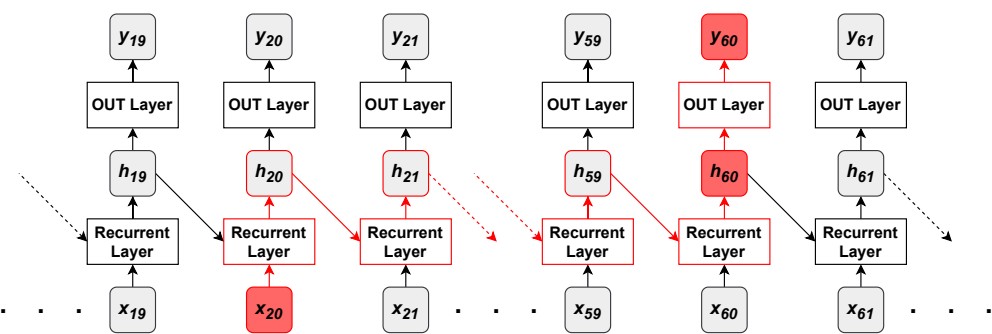

**Figure 8.** The red path in the recurrent model denotes the flow that information about an input sample ($x_{20}$) must follow before reaching an output layer ($y_{60}$) that might require it.

Supposing that the output element at position $t = 60$ has a strong dependency on the input at position $t = 20$, information about the input sample $x_{20}$ is useful to predict the output sample $y_{60}$. The output sample $y_t$ is predicted starting from $h_t$, a lossy summary of the past inputs yielded by the recurrent layer; hence, the only way for the output layer to know about $x_{20}$ is through $h_{60}$. The recurrent layer first captures information about $x_{20}$ through $h_{20}$, which has to pass by many steps and then aggregate information about many other input elements, before achieving $h_{60}$. There is no guarantee that after so many recurrent steps, adequate information about $x_{20}$ is preserved into $h_{60}$. In fact, $h_{60}$ may just contain information about the most recent samples and have no information about $x_{20}$ at all. The short-term memory of recurrent networks is one of their major drawbacks and one of the main reasons why attention mechanisms and Transformers were originally introduced in deep learning (see Section 4.1).

### 3.3. Hybrids and Variants of Deep Neural Networks

In recent years, specific deep neural networks have been proposed as *hybrids* or *variants* of the aforementioned architectures. Hybrid models combine multiple statistical or machine learning methods to improve the robustness and accuracy of forecasting. Towards the same goal are the works that propose variants of deep neural architectures properly adapted for time series forecasting tasks. Hybrids and variants of deep neural networks share the same limitations as the models they are based on. The most successful approaches are summarised in Table 7.

**Table 7.** Applications in time series forecasting using variants and hybrids of common deep neural networks. The symbol '+' denotes a combination of multiple models or methodologies. GARCH and ANN acronym stand for *Generalized AutoRegressive Conditional Heteroskedasticity* [105] and *Artificial Neural Networks*, respectively.

| Ref. | Year | Architecture | Application |
|---|---|---|---|
| [106] | 2016 | Autoencoder + LSTM | Solar power |
| [107] | 2017 | Autoencoder + LSTM | Stock prices |
| [108] | 2017 | CNN + LSTM | Stock prices |
| [109] | 2018 | CNN + LSTM | Electricity prices |
| [110] | 2018 | CNN + LSTM | Electricity load |
| [111] | 2018 | CNN + LSTM | Wind speed |
| [112] | 2018 | LSTM + Attention mechanism (see Section 4.1.1) | Stock market |
| [113] | 2018 | LSTM + GRU | Stock prices |
| [114] | 2018 | GARCH + LSTM | Stock prices |
| [115] | 2018 | GRU variant | Traffic forecasting |
| [116] | 2018 | CNN + LSTM | $PM_{2.5}$ concentration |
| [117] | 2018 | ANN + LSTM + CNN | $PM_{2.5}$ concentration |
| [118] | 2019 | LSTM + Attention mechanism (see Section 4.1.1) | Online Sales/Electricity prices |
| [119] | 2019 | LSTM + Attention mechanism (see Section 4.1.1) | Solar generation |
| [120] | 2019 | LSTM + Attention mechanism (see Section 4.1.1) | Electricity load |
| [27] | 2019 | CNN + Attention mechanism (see Section 4.1.1) | Traffic/Stock market |
| [121] | 2020 | CNN + LSTM | Stock market/Temperature |
| [122] | 2020 | LSTM + Fuzzy Logic | COVID-19 |
| [23] | 2020 | TCN + Attention | Remaining Useful Life |
| [123] | 2023 | TCN + LSTM/GRU | Chaotic Time Series/ECG |

*3.4. Graph Neural Networks*

A recent promising research direction in multivariate time series forecasting is the application of *Graph Neural Networks* (GNNs) [124,125]. The original domain of GNNs is the handling of *spatial dependencies* among entities in a graph-modelled problem. In detail, GNNs can be used to generate representations of such entities that depend on the structure of the graph and on any additional information. A graph $\mathcal{G}'$ is formally defined as a tuple $\mathcal{G}' = [\mathcal{V}, \mathcal{E}]$, where $\mathcal{V}$ denotes the set of *nodes* and $\mathcal{E}$ is the set of *edges*, the connections between the nodes of the graph, represented, in this case, with an *adjacency matrix*. The definition of this matrix is based on a *metric function* that can be either *a priori* fixed or *learned* during the training process [125]. The basic idea of a GNN can be summarised by the use of three main operators: *aggregate*, *combine*, and *readout*. The $k$-th GNN layer performs the aggregate and combines operators. The former consists of agglomerating, for each graph node $v \in \mathcal{V}$, information from its neighbourhood $N(v)$ as defined by the adjacency matrix:

$$\vec{h}_{N(v)}^k = aggregate(\vec{h}_u^{k-1} : u \in N(v)) \quad \forall k > 1 \tag{19}$$

where $\vec{h}_u^{k-1}$ is the feature vector of the $u$-th neighbouring node of $v$, yielded by the previous GNN layer $k - 1$, and $\vec{h}_{N(v)}^k$ is the aggregated information. The latter combines the aggregated information $\vec{h}_{N(v)}^k$ with the feature vector $\vec{h}_v^{k-1}$ of the current node $v$:

$$\vec{h}_v^k = combine(\vec{h}_v^{k-1}, \vec{h}_{N(v)}^k) \quad \forall k > 1. \tag{20}$$

When $k = 1$ the *aggregate* operator is not defined, whereas the *combine* operator reduces to:

$$\vec{h}_v^1 = \vec{x}_v, \tag{21}$$

where $\vec{x}_v$ represents the input feature vector associated to the $v$-th node. Furthermore, the readout operator is applied on the output of the last GNN layer $\mathcal{K}$ in order to obtain a final vector $\vec{h}_G$ representing the whole graph $\mathcal{G}' = [\mathcal{V}, \mathcal{E}]$:

$$\vec{h}_G = readout(\vec{h}_v^{\mathcal{K}} : v \in \mathcal{V}) \tag{22}$$

In the case of multivariate time series forecasting, GNNs have been successfully applied as feature selection mechanisms. It is important to remark that GNNs could also be applied to *spatio-temporal time series forecasting* which is not the object of the survey. GNN time series forecasting applications are described in Table 8.

**Table 8.** GNN applications on time series forecasting.

| Ref. | Year | Application |
|------|------|-------------|
| [28] | 2020 | Traffic/Electricity load/Exchange rate |
| [29] | 2021 | Solar energy/Traffic/Electricity load/Exchange rate |
| [126] | 2022 | Stock market |
| [127] | 2022 | $PM_{2.5}$/Traffic/Wind speed |
| [128] | 2022 | Stock market |
| [129] | 2022 | Electricity load/Solar energy/Traffic |
| [21] | 2022 | Solar energy/Wind power generation/Electricity load/Exchange rate |
| [130] | 2022 | Solar energy/Traffic/Electricity load/Exchange rate |
| [22] | 2023 | Solar energy/Traffic/Electricity load/Exchange rate |

*3.5. Deep Gaussian Processes*

Let $D = (\vec{x}_1, \vec{x}_2, \ldots, \vec{x}_n)$ be a data set and $Y = (\vec{y}_1, \vec{y}_2, \ldots, \vec{y}_n)$ the target output, a *Gaussian Process* [131] is a Bayesian model composed of a collection of random variables, any finite number of which have a joint Gaussian distribution, and it is fully defined by a mean function $m(\vec{x}_i)$ and covariance function $k(\vec{x}_i, \vec{x}_j)$, which is usually a *Mercer kernel* [131,132]. The analytical solution of a Gaussian Process entails computing the inverse of the covariance matrix of observations, which has a computational complexity of $O(n^3)$. A common approach for coping with this computational drawback is the use of *Sparse Gaussian Process* [133]. This method consists of considering a reduced set of $m$ ($m \ll n$) training samples, the so-called *inducing variables*, reducing, in this way, the complexity to $O(nm^2)$. A sequence of Gaussian Processes defines a Bayesian model called *Deep Gaussian Process* (DGP) [134]. As shown in Figure 9, in DGPs the output of the single Gaussian Process located at the previous layer is fed as an input to the Gaussian Process located at the next layer. Unlike the rest of the deep learning techniques, Deep Gaussian Processes can estimate not only the value of future time series samples but also provide a confidence interval of the predictive time series value representing, in this way, the uncertainty of the model. DGP time series forecasting applications are described in detail in Table 9.

**Table 9.** DGP applications on time series forecasting.

| Ref. | Year | Application |
|------|------|-------------|
| [135] | 2017 | Crop Yield forecasting |
| [136] | 2020 | Crop Yield forecasting |
| [137] | 2022 | Electricity load |
| [138] | 2023 | Car-hailing demand |
| [139] | 2023 | Ozone concentration forecasting |

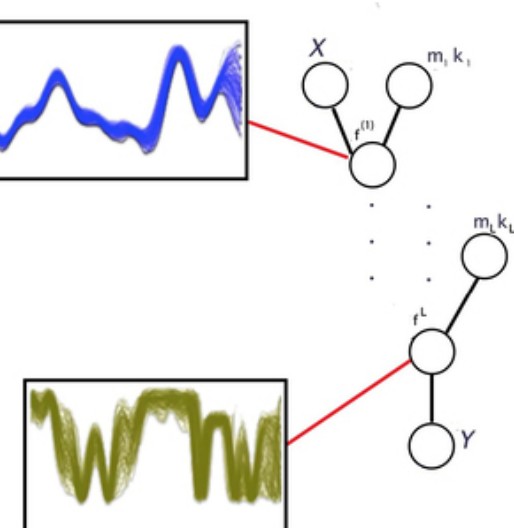

**Figure 9.** Deep Gaussian Processes. In the figure $\mathcal{X}$, $\mathcal{Y}$ represent the data set and the desired output, respectively, $f^{(i)}$ is the latent function to be estimated and $m_i$, $k_i$ represent the mean and covariance function of $i$-th layer.

### 3.6. Generative Models

Among the latest trends in deep learning research, there are the so-called *generative models*, specifically *Generative Adversarial Networks* (GANs) and *Diffusion Models* (DMs). Both families are popular for their groundbreaking capabilities in generating synthetic images. These successes encouraged researchers to apply GANs and DMs to sequential data as well, including time series. As generative models, both methodologies have been used for *time series generation* tasks. However, it can adapt them for other time series-related tasks as well, specifically *time series forecasting*. Sections 3.6.1 and 3.6.2 provide an overview of GANs and their usage for time series forecasting, respectively; Sections 3.6.3 and 3.6.4 discuss diffusion models and their applications to time series forecasting, respectively.

#### 3.6.1. Generative Adversarial Networks

A GAN [140] is composed of two separate artificial neural networks: a *generator network G* that generates synthetic data, with the same distribution of the input ones, and a *discriminator network D* that classifies input data as either real or synthetic. $G$ and $D$ are trained with an *adversarial training* approach. $G$ takes random noise as input and it has to transform the noise into a synthetic data sample following the same distribution of the real data. $D$ receives both real and generated samples and it estimates the probability that any given data sample comes from the real data rather than from the generator. The two networks are trained jointly with a *minimax two-player game* [140], i.e., the discriminator is trained to maximise the correct classification ratio for both real and generated samples. Whereas, the generator has the goal to trick the discriminator into misclassifying generated samples by minimising the correct classification ratio. This training procedure is expressed by the objective function:

$$\min_{G} \max_{D} V(D, G) = \mathbb{E}_{\vec{x} \sim p_x(\vec{x})}[\log D(\vec{x})] + \mathbb{E}_{\vec{z} \sim p_z(\vec{z})}[\log(1 - D(G(\vec{z})))]. \tag{23}$$

where, $\vec{x}$ is a real data point, sampled from the real data distribution $p_x(\vec{x})$; $\vec{z}$ is a noise vector, sampled from a distribution $p_z(\vec{z})$, *a priori* fixed; $D(\vec{x})$ is the probability distribution estimated by the discriminator; $G(\vec{z})$ is the sample produced by the generator, starting from the noise $\vec{z}$. GANs can be implemented by any neural architecture for the generator and the discriminator. For instance, $G$ and $D$ can be implemented by MLPs [67], as originally proposed in [140], CNNs (see Section 3.1), with some architectural constraints to stabilise the training procedure [141], or LSTM (see Section 3.2.4) networks [142].

### 3.6.2. Generative Adversarial Networks in Time Series Forecasting

In literature, there are two main approaches for using GANs in time series forecasting: as *data augmentation* or as an *end-to-end forecasting model*. In the former case, GANs generate synthetic time series in a given domain (e.g., financial or health-related time series) in order to augment the original small data set. The augmented data set, with both real and generated time series, is then used to train a traditional forecasting model, e.g., a model based on an LSTM network. In the latter case, the forecasting model is a GAN itself, and it generates future samples starting from previous ones and, eventually, other *exogenous inputs* (In time series forecasting, a variable is said to be exogenous if it is not a time series sample, but it can still affect the time series samples. For instance, temperature may be an *exogenous* variable in rainfall time series forecasting) [143]. For example, in [144] a GAN is firstly used to augment a building energy consumption data set and then, an ensemble of five traditional predictive models is trained on such augmented data set. In particular, to augment the data set, *Conditional GANs* (CGANs) [145] and *Information Maximising GANs* (InfoGANs) [146] are compared with each other. Similarly, in [147] COVID-19 epidemic data is generated by a custom GAN with an LSTM generator and an MLP discriminator. Furthermore, a Transformer is used to make a forecasting model trained on the GAN-augmented data set. Moreover, some GAN-based models have been specifically developed for time series generation, e.g., *QuantGAN* [148], for financial time series with long-term time dependencies, *SynSigGAN* [149], for continuous biomedical signals, *Recurrent Conditional GANs* (RCGANs) [150], for medical data, *TimeGAN* [151], a framework for domain-agnostic time series generation, *Conditional Sig-Wasserstein GAN* (Sig-WCGAN) [152], and *TTS-GAN* [153], entirely based on Transformers. Some of the aforementioned GANs, e.g., RCGAN, TimeGAN, Sig-WCGAN, are *conditional* GANs [145], i.e, time series are not generated from only random noise but also conditioned on the real time series and/or related information, e.g., exogenous inputs, for improving generated time series quality. The use of conditional GANs is popular for end-to-end forecasting, where the generated time series window, typically in the short-term future, is often conditioned on previous samples and on other exogenous inputs (see [154–158]). Table 10 collects some works on GAN applications for time series forecasting.

**Table 10.** GAN applications on time series forecasting.

| Ref. | Year | Application |
|------|------|-------------|
| [159] | 2018 | Stock market |
| [160] | 2019 | Traffic forecasting |
| [154] | 2019 | Lorenz/Mackey-Glass/Internet Traffic data |
| [161] | 2019 | Medicine expenditure |
| [162] | 2019 | Electricity load |
| [163] | 2020 | Stock price |
| [164] | 2020 | Long-term benchmark data sets (see Section 6.2) |
| [165] | 2020 | Soil temperature |
| [166] | 2021 | Stock market/Energy production/EEG/Air quality |
| [156] | 2021 | Internet Traffic data |
| [167] | 2021 | Store Item Demand/Internet Traffic/Meteorological data |
| [168] | 2021 | Wind power/Solar power |
| [144] | 2021 | Energy consumption |
| [169] | 2021 | Electricity load |
| [170] | 2022 | Trajectories forecasting |
| [147,155] | 2022 | COVID-19 |
| [157,158] | 2022 | Photovoltaic power |
| [171] | 2022 | Building power demand |
| [172] | 2023 | Financial time series |

### 3.6.3. Diffusion Models

A new family of generative architectures are the *diffusion models*. They have been showing cutting-edge performance in generating samples that reflect the observed data in different domains, e.g., image generation, video generation, and text synthesis. Three key formulations are used to develop diffusion-based approaches for short-term time series applications: denoising diffusion probabilistic models (*DDPMs*) [173,174], score-based generative models (*SGMs*) [175], and stochastic differential equations (*SDEs*) [176]. Diffusion models aim to approximate a generative process $\mathcal{G}$ that generates new samples drawn from an *underlying distribution* $q(x)$, given some observed data $x$ from the same distribution. To approximate $\mathcal{G}$, in the forward process, a progressive injection of Gaussian noises on the observed data is performed by the *majority* of diffusion models. Furthermore, a reverse process is applied, by a learnable transition kernel, to reconstruct the original data. Most diffusion models assume that, after a finite number of noise injection steps, $q(x)$ distribution of the observed data will become a Gaussian distribution. Therefore, the goal of diffusion models is to find the *probabilistic process* $\mathcal{P}$ that approximates $q(x)$ distribution of original data from the Gaussian distribution. In this manner, any sample of Gaussian distribution can be transformed by $\mathcal{P}$ into a new sample of $q(x)$ distribution of observed data $x$. The principle of diffusion models is to progressively perturb the observed data $x$ with a random noise by a *forward diffusion process* $\mathcal{F}$ before recovering the original data using a *backward reverse diffusion process* $\mathcal{B}$. A deep neural network is used in the $\mathcal{B}$ process to approximate the amount of noise that must be removed in the denoising steps to recover the original data. For the sake of readability, the theoretical foundations of diffusion models and their main architectures are omitted in the section and moved in Appendix B, whereas the diffusion models for short-term time series forecasting are described in the following subsection.

### 3.6.4. Diffusion Models in Short-Term Time Series Forecasting

In recent years, several diffusion-based approaches for time series forecasting have been proposed. They are based on the three predominant methods of diffusion model described in Appendix B. The first prominent diffusion model architecture for time series forecasting is *TimeGrad* [177], which is a DDPM variant. The forward process of TimeGrad injects noises into data at each predictive sample, and then denoises gradually through the backward process conditioned on previous time series samples. For encoding the previous time series samples, TimeGrad uses an RNN architecture, e.g., LSTM (see Section 3.2.4) or GRU (see Section 3.2.5), The objective function of TimeGrad is represented by a negative log-likelihood, denoted as:

$$\sum_{t=t_0}^{T} -\log p_\theta(x_t^0 | h_{t-1}), \tag{24}$$

where $[t_0, T]$ is the *prediction length*. The Equation (24) can be reformulated considering the lower bound:

$$\mathbb{E}_{k,x_t^0,\epsilon}[\delta(k) || \epsilon - \epsilon_\theta(\sqrt{\tilde{a}_k} x_t^0 + \sqrt{1 - \tilde{a}_k}\epsilon, h_{t-1}, k)||^2]. \tag{25}$$

The parameters $\theta$ are estimated during the training, minimising the negative log-likelihood objective function with a stochastic sampling. Furthermore, future time series samples are generated with a step-by-step procedure. The observation for the next samples at time $T + 1$ is predicted in a similar way as *DDPM* (see Appendix B). Similarly, the *ScoreGrad* model [178], based on the same target distribution of TimeGrad, defines a continuous diffusion process using *SDEs* (see Appendix B). ScoreGrad consists of two modules: the former is a *feature extraction module* (e.g., an RNN) almost identical to TimeGrad, or an attention-based network, e.g., Transformer (see Section 4.1), for computing the hidden state

$h_t$ of previous time samples, the latter is a conditional SDE-based score-matching module. The objective function of ScoreGrad is computed as follows:

$$\sum_{t=t_0}^{T} L_t(\theta),\tag{26}$$

with $L_t(\theta)$ being:

$$\mathbb{E}_{k,x_t^0,x_t^k}[\delta(k)||s_\theta(x_t^k,h_t,k) \,-\, \nabla_{x_t^k}\log q_{ok}(x_t|x_t^0)||^2].\tag{27}$$

It is worthwhile to remark that, in Equation (27), the general formulation of SDEs has been used for the sake of simplicity. Recently, time series research has paid attention to avoiding *model overfitting phenomena* in the forecasting of short time series. $D^3VAE$ [179], tries to address the problem of short time series, applying a coupled diffusion process for *time series data augmentation*, and then it performs a *bidirectional autoencoder* (BVAE), as a *score model*. Moreover, $D^3VAE$ takes into account the decoupling of latent variables by reducing total correlation to improve prediction interpretability and stability. Furthermore, the objective function of $D^3VAE$ includes the *mean square error* (MSE), which highlights the requirement of supervision, among the forecast and current samples in the prediction window. Unlike TimeGrad, $D^3VAE$ injects noises separately into the previous samples (*context*) $x_{1:t_0-1}^0$ and the *prediction window* $x_{t_0:T}^0$ by the coupled diffusion process:

$$x_{1:t_0-1}^k = \sqrt{\tilde{a}_k}x_{1:t_0-1}^0 \,+\, \sqrt{1-\tilde{a}_k}\epsilon,\tag{28}$$

$$x_{t_0:T}^k = \sqrt{\tilde{a}'_k}x_{t_0:T}^0 \,+\, \sqrt{1-\tilde{a}'_k}\epsilon,\tag{29}$$

where $\epsilon$ indicates the standard Gaussian noise. Short time series forecasting benefits the simultaneous improvement of the context and prediction window provided by the diffusion process. The $\mathcal{B}$ process is made up of two steps. The former forecasts $x_{t_0:T}^k$ with a BVAE model, considering the context $x_{1:t_0-1}^k$. The latter denoises the output $\tilde{x}_{t_0:T}^k$ of BVAE with a denoising score matching module, as follows:

$$x_{t_0:T}^k \leftarrow \tilde{x}_{t_0:T}^k - \sigma_0^2 \nabla_{\tilde{x}_{t_0:T}^k} E(\tilde{x}_{t_0:T}^k;e),\tag{30}$$

where $E(\tilde{x}_{t_0:T}^k;e)$ is the energy function. The objective function of $D^3VAE$ is composed of four losses, that can be written as follows:

$$\lambda_1 D_{KL}(q(x_{t_0:T}^k||p_\theta(\tilde{x}_{t_0:T}^k)) \,+\, \lambda_2 L_{DSM} \,+\, \lambda_3 L_{TC} \,+\, L_{MSE},\tag{31}$$

where $\lambda_1, \lambda_2, \lambda_3$ are the regularisation parameters of *divergence between target distribution and distribution of prediction window*, *denoising score matching objective*, and *total correlation among latent variables*, respectively. Diffusion models for time series forecasting are summarised in Table 11.

**Table 11.** Recent diffusion models for time series forecasting.

| Ref. | Year | Model |
|------|------|-------|
| [177] | 2021 | TimeGrad |
| [178] | 2021 | ScoreGrad |
| [180] | 2022 | DSPD |
| [179] | 2022 | $D^3VAE$ |

## 4. Deep Learning Models for Long-Term Forecasting

In long-term forecasting, the skeleton of a time series can be approximated by using the *Transfomer* architecture. Firstly, the original Transformer architecture (Section 4.1) is

described and attention mechanisms are presented (Sections 4.1.1 and 4.1.2). Furthermore, the main limitations of Transformers are discussed (Section 4.1.3) and *variants of Transformer*, properly designed to cope with long-term time series forecasting tasks, e.g., *Informer*, *Autoformer*, *FEDFormer* and *Crossformer*, are presented (Section 4.1.4).

*4.1. Transformers*

The Transformer [181] is a deep-learning architecture borrowed from Natural Language Processing. It can be described as: "*a model architecture eschewing recurrence and relying entirely on attention mechanisms to draw global dependencies between input and output*" [181]. The Transformer architecture was proposed to overcome the main drawbacks of recurrent models (see Sections 3.2.2 and 3.2.6) with sequence modelling tasks:

1. The output state $h_t$ of a recurrent layer at time $t$ depends on the state $h_{t-1}$, produced at the previous time step. This inherent sequential nature prohibits the intra-sequence parallelism of recurrent networks.
2. Recurrent networks cannot generally learn relationships between sequences of distant samples, since information must first pass through all data samples in between (see Figure 8).

The standard Transformer follows a general *encoder-decoder* architecture for sequence-to-sequence transduction, as shown in Figure 10.

In time series forecasting, the Transformer's input is a time-ordered sequence of past samples $\mathcal{X} = [\vec{x}_1, \vec{x}_2, \dots, \vec{x}_L]$ where $L$ is the *sequence length* and $\vec{x}_i \in \mathbb{R}^d$ is the $i$-th sample of a $d$-dimensional multivariate time series. Due to the use of *attention mechanisms*, Transformers make no assumption on any intrinsic temporal or spatial ordering of input elements, namely inputs are seen as a *set of samples* rather than ordered sequences of samples. If there is a relevant input ordering for the modelling task, e.g., time series forecasting, the ordering should be encoded in the input embedding. In Transformers, this is commonly achieved by summing a *positional embedding* $E_{pos}$ to the main sample embedding $F(\mathcal{X})$ [181]:

$$input = F(\mathcal{X}) + E_{pos} \tag{32}$$

where the matrix (Differently from what appears in some machine learning papers, the more precise tensor product notation is used in the whole work for representing matrices) $F(\mathcal{X}) \in \mathbb{R}^L \otimes \mathbb{R}^D$ represents a *projection* of the input sequence in a higher $D$-dimensional space ($D > d$). In time series forecasting, a *1D convolutional layer* is commonly used with $D$ learned kernels, as described in Section 3.1, in order to extract a $D$-dimensional representation for each sample in $\mathcal{X}$ [181–184]. $E_{pos}$ can either be a learned embedding or a fixed embedding. A naive solution, yet effective, consists of using a *sinusoidal position encoding* [181]. However, in time series forecasting, other positional embeddings can be used as well, e.g., *temporal-based embeddings* [182–184]. The encoder and the decoder can have two different separated embeddings, or they can share the same embedding if input and output samples belong to the same set. In time series forecasting, the *encoder input* is the complete sequence of past samples $\mathcal{X}$, while the *decoder input* is commonly composed of the most recent part of $\mathcal{X}$ (e.g., the second half of $\mathcal{X}$, i.e., $[\vec{x}_{L/2}, \vec{x}_{L/2+1}, \dots, \vec{x}_L]$) and a *zero-vector* whose length is equal to prediction length $P$, see Equation (1). The encoder and decoder are composed of $N_e$ and $N_d$ stacked layers, respectively (see Figure 10). The output of a layer is the input for the next layer. Each encoder layer has two sublayers: a *self-attention* layer, that relates each input sample with the rest of the samples, and a *shallow feed-forward dense* layer, shared along the sequence axis, that works as a nonlinear projection layer. To foster gradient propagation and training, each sublayer's input is added to its own output with a *residual connection* [185], and *layer normalization* [186] is used to normalise the samples of the resulting sequence into a normal distribution with a learned mean and standard deviation. Each decoder layer follows the same overall structure of a generic encoder layer, but it has one additional sublayer. The first sublayer implements a particular kind of *self-attention* mechanism, the so-called *causal* (or *masked*) *self-attention* [181]. It works



similarly to the encoder layer's self-attention, as each input sample is related to the others in the decoder's input sequence, but it also uses *masking* to prevent future samples from being considered when the processing of the current sample occurs. Furthermore, the output of the causal self-attention sublayer is related to the encoder's hidden representation (that is, the output of the final encoder layer) by a *cross-attention* layer. As in encoder layers, a *Multi-Layer Perceptron* [67] with one hidden layer is used for projecting nonlinearly the output of cross-attention. Moreover, each sublayer is wrapped by a residual connection followed by layer normalization. Finally, the output of the last decoder layer is fed to a *final prediction layer*.

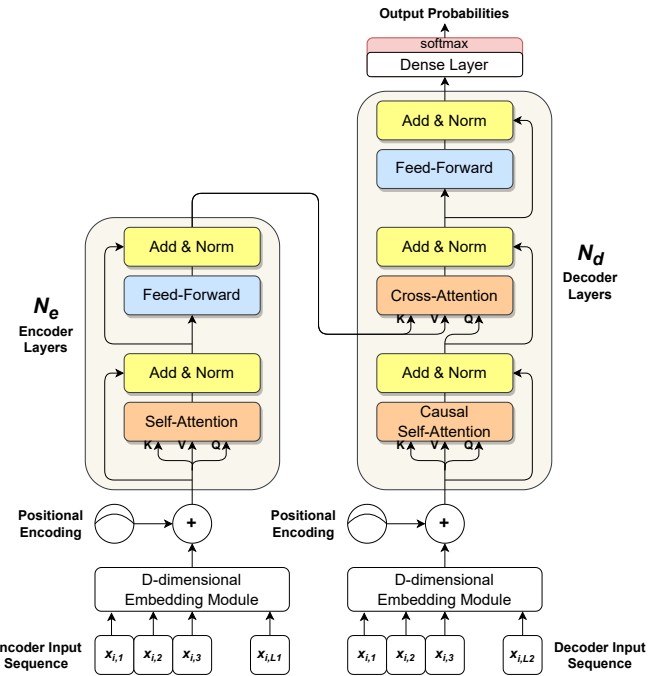

**Figure 10.** Transformer architecture. On the left side, the *encoder* processes an input sequence, producing a hidden representation. On the right side, the *decoder* uses the encoder's output to generate the output sequence. The decoder works in an autoregressive way, consuming past generated samples as additional inputs to generate the next output sample.

### 4.1.1. Attention Mechanisms

The most important computational blocks of a Transformer are *attention mechanisms*, that allow the model to *focus its attention* on specific parts of the input, depending on the information being processed. Among various definitions of attention, Transformers adopt the so-called *scaled dot-product attention*, which is very similar to *multiplicative attention* [187]. Attention mechanisms operate on the following elements: a set of *queries* $Q \in \mathbb{R}^M \otimes \mathbb{R}^{D_k}$ that represents the information being processed by the model, and sets of *keys* $K \in \mathbb{R}^N \otimes \mathbb{R}^{D_k}$ and *values* $V \in \mathbb{R}^N \otimes \mathbb{R}^{D_v}$, where $D^k$ and $D^v$ denote the dimension of space where queries, keys and values are projected. Moreover, $N$ denotes the cardinality of both keys and values, while $M$ is the cardinality of the input queries. The output $Y$ for all queries is computed as follows:

$$Y = \text{Attention}(K, V, Q) = \text{softmax}\left(\frac{QK^\top}{\sqrt{D_k}}\right)V \tag{33}$$

The attention output $Y \in \mathbb{R}^M \otimes \mathbb{R}^{D_v}$ is a matrix whose *i*-th row contains the output vector for the *i*-th query. Note that the softmax in Equation (33) is applied row-wise to its input matrix. *Where do these queries, keys and values come from?* First of all, keys and values are often the same vectors, i.e., a value vector coincides with its key. Furthermore, as described in Section 4.1, the Transformer performs attention in two ways, *self-attention* and *cross-attention*.

In self-attention, queries and values are the same vectors; in cross-attention queries come from the previous decoder sublayer, while key and value vectors are given by the encoder's hidden representation.

### 4.1.2. Multi-Head Attention

*Multi-Head Attention* (MHA) is a more advanced version of the aforementioned scaled dot-product attention. As discussed in [181], the scaled dot-product attention permits a network to attend over a sequence. However, often there are multiple different aspects a sequence element wants to attend to, and a single weighted average is not an adequate option for it. This motivates the extension of the scaled dot-product attention to MHA, which allows the model to jointly attend to information from diverse representation subspaces, as shown in Figure 11.

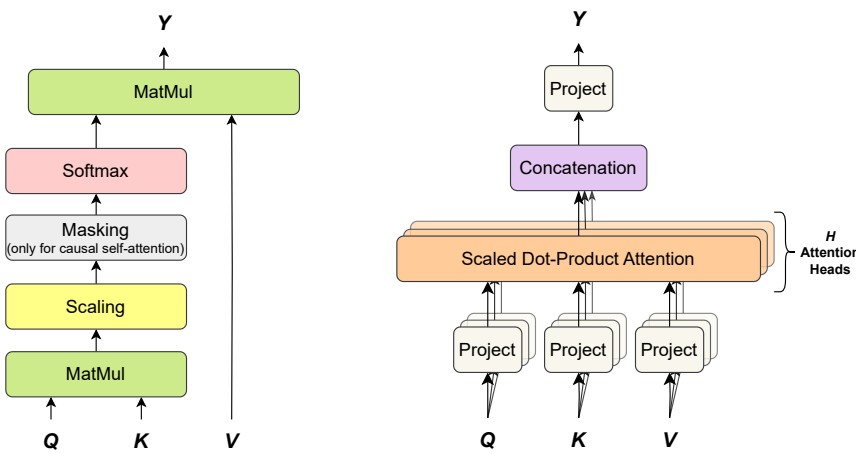

(**a**) Scaled dot-product attention.   (**b**) Multi-Head Attention with *H* heads.

**Figure 11.** Comparison of a single scaled dot-product attention (**a**) and multi-head attention with *H* attention heads (**b**).

In MHA, keys, values and queries are linearly projected *H* separate times, by three learned projection matrices, onto spaces of dimensions $D_k$, $D_v$ and $D_k$ respectively. Furthermore, a scaled dot-product attention is applied to each of these projections and the results are concatenated together and re-projected onto the previous layer space. Each projection-attention pair defines a so-called *attention head* $h_i$. For the sake of simplicity, keys, values and queries are assumed to have the same dimension *D*. Each attention head $h_i$ has three learned matrices: $W_i^K \in \mathbb{R}^D \otimes \mathbb{R}^{D_k}$, $W_i^V \in \mathbb{R}^D \otimes \mathbb{R}^{D_v}$ and $W_i^Q \in \mathbb{R}^D \otimes \mathbb{R}^{D_k}$, used to project keys, values and queries, respectively. Each attention head applies a scaled dot-product attention (see Equation (33)) to the projected keys, values and queries (see Section 4.1.1):

$$h_i = \text{Attention}(KW_i^K, VW_i^V, QW_i^Q) \quad \forall i \in [1, H] \tag{34}$$

Finally, the attention output *Y* is given by:

$$Y = \text{MHA}(K, V, Q) = \text{Concatenate}(h_1, h_2, \ldots, h_H)W^o \tag{35}$$

where the outputs $h_i$ from all attention heads are concatenated into a single $\mathbb{R}^M \otimes \mathbb{R}^{HD_v}$ matrix and then re-projected linearly to the original *D*-dimensional space via an additional projection matrix $W^o \in \mathbb{R}^{HD_v} \otimes \mathbb{R}^D$.

### 4.1.3. Shortcomings of Transformers

There are three main shortcomings of Transformers. Firstly, Transformers are *locally-agnostic*, that is, the scaled dot-product of the attention mechanism (see Equation (33)) is insensitive to the local context, which can make the model prone to anomalies in time

series forecasting [188]. Furthermore, Transformers suffer of *memory bottleneck*, i.e., Transformers' space complexity is $\mathcal{O}(L^2)$ with sequence length $L$ [188]. Similarly, Transformers also have the same time complexity, limiting their application to the long-term forecasting. These shortcomings are faced by some of the Transformer variants described in the following section.

4.1.4. Transformer Variants for Time Series Forecasting

In recent years, many variants of the naive Transformer [181], specific for time series forecasting, have been proposed. Key innovations that these variants suggest concern the embedding layer, attention mechanisms and even the encoder-decoder structure. Most of the literature focused on the design of alternative attention mechanisms that are more suitable for time series forecasting tasks. One of the first such works is the *LogTrans* [188], which handles two limitations of the traditional Transformer: locally-agnostic and memory bottleneck (see Section 4.1.3). The former limitation is tackled using causal convolutions (see Section 3.1.2) to generate keys and queries in the self-attention module. For the latter, a *log-sparse mask* is considered in order to reduce the computational complexity (see Section 4.1.3) of multi-head attention. Inspired by the idea of LogTrans, another variant, the *Informer* [182], defines a new sparse measure to characterise a subset of the most informative queries before applying attention. In addition, this strategy also allows for the reduction of the computational complexity of attention mechanisms. Unlike LogTrans and Informer, the *Autoformer* [183] replaces the standard scaled dot-product attention with an *autocorrelation mechanism*. Additionally, a *decomposition module* is employed to break down the time series into *trend* and *seasonal* components, assuming implicitly that they exist and are additive. The autocorrelation mechanism measures the time-delay similarity between input signals and aggregates the top-k similar sub-series to produce the output. The *FEDformer* [184], based on the work of *Linformer* [189], applies attention to a low-rank approximation of the input based on the *Restricted Isometry Property (RIP) matrix theory* [190]. First, it represents the input signal into a frequency domain (either Fourier or Wavelet). Furthermore, it achieves a linear complexity by applying simplified attention mechanisms on a randomly selected subset of frequencies with a fixed size $m$. Recently, research efforts have moved from attention mechanisms to *input representation*, specifically concerning how to relate the dimensions of a multivariate time series and how to project the input sequence into a latent representation. The *patchTST* [191] assumes *channel independence*, i.e., independence among the dimension of the input multivariate time series, processing each dimension as a univariate time series. PatchTST segments each input sequence into shorter, local sub-sequences that are fed as input samples to a naive Transformer encoder [181]. All time series dimensions are implicitly related via the *sharing* of the encoder weights. A similar consideration is adopted by the *Crossformer* [192], which segments each dimension of the input time series into non-overlapping shorter sub-sequences. Unlike patchTST, however, the Crossformer explicitly defines the relations among all dimensions using a *Two-Stage Attention* (TSA) mechanism. Furthermore, Crossformer follows a *Hierarchical Encoder-Decoder* architecture, in which multiple layers of TSA are used to capture relations at multiple time scales. Another relevant work is the *Pyraformer* [193], which proposes a *Pyramidal Attention Module* (PAM) to capture long-term dependencies while achieving a complexity that is linear in the sequence length. Essentially, PAM consists of applying the classic scaled dot-product attention in a sparse fashion according to a pyramidal graph, built using a cascade of strided convolutions, that defines a multi-scale representation of the input sequence. According to PAM, each node of the graph is a query and it can attend only those nodes (keys) that are its direct neighbours in the graph. In this way, Pyraformer is able to capture both short-term and long-term dependencies while still achieving a linear complexity. Similarly to Pyraformer, *Scaleformer* [194] addresses the importance of multi-scale dependencies in time series forecasting. The approach is orthogonal to many time series Transformers and, as such, it has been empirically evaluated with some of the aforementioned models like the *Autoformer* [183] and the *FEDformer* [184].

Given an input past sequence and the corresponding target sequence, the main idea is to apply one of the above-mentioned Transformer models, multiple times at multiple time scales. At a given scale $s_i$, the input to the encoder is the original look-back window but downsampled by a factor $s_i$ using average pooling; the input to the decoder is given by the model forecast at the previous scale $s_{i-1}$, but upsampled by a fixed factor $s$ through linear interpolation. To mitigate error propagation and distribution shifts that are due to repeated upsampling operations, the encoder and decoder's inputs are first normalised using *Cross-Scale Normalization*. Finally, a loss function, based on an adaptive loss, is applied at each time scale between the model forecast and the target sequence, which is also downsampled by a factor $s_i$ via average pooling. The *Triformer* [195] proposes a particular architecture that integrates attention mechanisms and recurrent units to ensure high efficiency and accuracy. The former is achieved by a *patch attention mechanism* with *linear complexity*; the latter is obtained by using *variable-specific parameters*. The patch attention mechanism splits the input sequence in $P$ patches of length $S$ and assigns a learnable *pseudo-timestamp* $T_p$ to each patch. When patch attention is applied, each *pseudo-timestamp* $T_p$ is considered as a query $Q$ for its patch only. Moreover, *variable-specific parameters* are introduced by factorising the projection matrices (i.e, $W^K$ and $W^V$) into three matrices: *left variable-agnostic* matrix $L \in \mathbb{R}^D \otimes \mathbb{R}^a$, *middle variable-specific* matrix $B \in \mathbb{R}^a \otimes \mathbb{R}^a$ and *right variable-agnostic* matrix $R \in \mathbb{R}^a \otimes \mathbb{R}^D$, where $a \ll D$. Finally, to cope with the limited temporal receptive field that is due to the patch mechanism, recurrent units are used to aggregate and control the information for all pseudo-timestamps of each layer before the final prediction. All above-mentioned variants of Transformer share the *over-stationarization problem* that consists in the inability to generate distinguishable attention scores when trained on stationarized series [196]. The *Non-stationary Transformer* [196] proposes a generic framework to overcome the problem of *over-stationarization*. This framework is composed of two interdependent modules: *Series Stationarization* and *De-stationary Attention*. The former attenuates the non-stationarity of the time series considered, using two sequential operations: *Normalization module*, which computes the mean and the variance for each input time series in order to transform it into a *stationary time series*, and a *De-normalization module*, which transforms the model outputs back into a *non-stationary time series*, using the mean and variance computed in the previous module. The latter is a novel attention mechanism, which can approximate the attention scores that are obtained without stationarization and discover the particular temporal dependencies from original non-stationary data. Transformer variants for time series forecasting are described in detail in Table 12. Further details on each Transformer variant, can be found in the original paper that presents the architecture.

**Table 12.** Recent variants of Transformer architecture for time series forecasting.

| Ref. | Year | Model |
|------|------|-------|
| [188] | 2019 | LogTrans |
| [182] | 2021 | Informer |
| [183] | 2021 | Autoformer |
| [184] | 2022 | FEDFormer |
| [193] | 2022 | Pyraformer |
| [195] | 2022 | Triformer |
| [196] | 2022 | Non-stationary Transfomers |
| [191] | 2023 | PatchTST |
| [192] | 2023 | Crossformer |
| [194] | 2023 | Scaleformer |

Table 13 reports an extensive comparison among all aforementioned Transformer variants. It has to be noted that, the reported results were collected from the original papers that tested a given model on a given data set (the reader can refer to the GitHub pages linked in the original papers of each architecture for reproducing the experiments, using the original experimental setups).

**Table 13.** Multivariate long-term forecasting benchmarks among variant of Transformer architectures with different prediction lengths $P \in [96, 192, 336, 720]$. The input length considered for ILI data set is 36 and 96 for the others. A lower MSE or MAE indicates a better prediction. The best results, for each data sets, are highlighted in bold.

| Models | | Crossformer | | PatchTST | | Non-Stationary | | Pyraformer | | FEDFormer | | Autoformer | | Informer | | LogTrans | | LSTM | | TCN | |
|---|---|---|---|---|---|---|---|---|---|---|---|---|---|---|---|---|---|---|---|---|---|
| Metric | | MSE | MAE | MSE | MAE | MSE | MAE | MSE | MAE | MSE | MAE | MSE | MAE | MSE | MAE | MSE | MAE | MSE | MAE | MSE | MAE |
| Weather | 96 | - | - | **0.149** | **0.198** | 0.173 | 0.223 | 0.354 | 0.392 | 0.217 | 0.296 | 0.266 | 0.336 | 0.300 | 0.384 | 0.458 | 0.490 | 0.369 | 0.406 | 0.615 | 0.589 |
| | 192 | - | - | **0.194** | **0.241** | 0.245 | 0.285 | 0.673 | 0.597 | 0.276 | 0.336 | 0.307 | 0.367 | 0.598 | 0.544 | 0.658 | 0.589 | 0.416 | 0.435 | 0.629 | 0.600 |
| | 336 | 0.495 | 0.515 | **0.245** | **0.282** | 0.321 | 0.338 | 0.634 | 0.592 | 0.339 | 0.380 | 0.359 | 0.395 | 0.578 | 0.523 | 0.797 | 0.652 | 0.455 | 0.454 | 0.639 | 0.608 |
| | 720 | 0.526 | 0.542 | **0.314** | **0.334** | 0.414 | 0.410 | 0.942 | 0.723 | 0.403 | 0.482 | 0.419 | 0.428 | 1.059 | 0.741 | 0.869 | 0.675 | 0.535 | 0.520 | 0.639 | 0.610 |
| Traffic | 96 | - | - | **0.360** | **0.249** | 0.612 | 0.338 | 0.684 | 0.393 | 0.562 | 0.349 | 0.613 | 0.388 | 0.719 | 0.391 | 0.684 | 0.384 | 0.843 | 0.453 | 1.438 | 0.784 |
| | 192 | - | - | **0.379** | **0.256** | 0.613 | 0.340 | 0.692 | 0.394 | 0.562 | 0.346 | 0.616 | 0.382 | 0.696 | 0.379 | 0.685 | 0.390 | 0.847 | 0.453 | 1.463 | 0.794 |
| | 336 | 0.530 | 0.300 | **0.392** | **0.264** | 0.618 | 0.328 | 0.699 | 0.396 | 0.570 | 0.323 | 0.622 | 0.337 | 0.777 | 0.420 | 0.733 | 0.408 | 0.853 | 0.455 | 1.479 | 0.799 |
| | 720 | 0.573 | 0.313 | **0.432** | **0.286** | 0.653 | 0.355 | 0.712 | 0.404 | 0.596 | 0.368 | 0.660 | 0.408 | 0.864 | 0.472 | 0.717 | 0.396 | 0.500 | 0.805 | 1.499 | 0.804 |
| Electricity | 96 | - | - | **0.129** | **0.222** | 0.169 | 0.273 | 0.498 | 0.299 | 0.183 | 0.297 | 0.201 | 0.317 | 0.274 | 0.368 | 0.258 | 0.357 | 0.375 | 0.437 | 0.985 | 0.813 |
| | 192 | - | - | **0.147** | **0.240** | 0.182 | 0.286 | 0.828 | 0.312 | 0.195 | 0.308 | 0.222 | 0.334 | 0.296 | 0.386 | 0.266 | 0.368 | 0.442 | 0.473 | 0.996 | 0.821 |
| | 336 | 0.323 | 0.369 | **0.163** | **0.159** | 0.200 | 0.304 | 1.476 | 0.326 | 0.212 | 0.313 | 0.231 | 0.338 | 0.300 | 0.394 | 0.280 | 0.380 | 0.439 | 0.473 | 1.000 | 0.824 |
| | 720 | 0.404 | 0.423 | **0.197** | **0.290** | 0.222 | 0.321 | 4.090 | 0.372 | 0.231 | 0.343 | 0.254 | 0.361 | 0.373 | 0.439 | 0.283 | 0.376 | 0.980 | 0.814 | 1.438 | 0.784 |
| ILI | 24 | 3.041 | 1.186 | **1.319** | **0.754** | 2.294 | 0.945 | 5.800 | 1.693 | 2.203 | 0.963 | 3.483 | 1.287 | 5.764 | 1.677 | 4.480 | 1.444 | 5.914 | 1.734 | 6.624 | 1.830 |
| | 36 | 3.406 | 1.232 | **1.579** | 0.870 | 1.825 | **0.848** | 6.043 | 1.733 | 2.272 | 0.976 | 3.103 | 1.148 | 4.755 | 1.467 | 4.799 | 1.467 | 6.631 | 1.845 | 6.858 | 1.879 |
| | 48 | 3.459 | 1.221 | **1.553** | **0.815** | 2.010 | 0.900 | 6.213 | 1.763 | 2.209 | 0.981 | 2.669 | 1.085 | 4.763 | 1.469 | 4.800 | 1.468 | 6.736 | 1.857 | 6.968 | 1.892 |
| | 60 | 3.640 | 1.305 | **1.470** | **0.788** | 2.178 | 0.963 | 6.531 | 1.814 | 2.545 | 1.061 | 2.770 | 1.125 | 5.264 | 1.564 | 5.278 | 1.560 | 6.870 | 1.879 | 7.127 | 1.918 |
| ETTm2 | 96 | - | - | **0.166** | **0.256** | 0.192 | 0.274 | 0.409 | 0.488 | 0.203 | 0.287 | 0.255 | 0.339 | 0.365 | 0.453 | 0.768 | 0.642 | 2.041 | 1.073 | 3.041 | 1.330 |
| | 192 | - | - | **0.223** | **0.296** | 0.280 | 0.339 | 0.673 | 0.641 | 0.269 | 0.328 | 0.281 | 0.340 | 0.533 | 0.563 | 0.989 | 0.757 | 2.249 | 1.112 | 3.072 | 1.339 |
| | 336 | - | - | **0.274** | **0.329** | 0.334 | 0.361 | 1.210 | 0.846 | 0.325 | 0.366 | 0.339 | 0.372 | 1.363 | 0.887 | 1.334 | 0.872 | 2.568 | 1.238 | 3.105 | 1.348 |
| | 720 | - | - | **0.362** | **0.385** | 0.417 | 0.413 | 4.044 | 1.526 | 0.421 | 0.415 | 0.422 | 0.419 | 3.379 | 1.388 | 3.048 | 1.328 | 2.720 | 1.287 | 3.153 | 1.354 |

## 5. Other Relevant Deep Learning Models

This section is reserved for those works that propose interesting architectures for short-term and long-term time series forecasting which do not fit the previously defined categories. Even though these models might share the same building blocks of well-known architectures (e.g., CNN, TCN, RNN, Transformer), due to their peculiarity and heterogeneity it has been decided to collect them under a proper section. In [197] a *Continuous Recurrent Unit* (CRU) based on stochastic differential equations (SDEs) and Kalman filters, that can handle irregularly sampled time series, is proposed. The *FiLM* (*Frequency-improved Legendre Memory*) model [198] generates a representation of past time series samples by *Legendre projections*. It uses a noise reduction module based on Fourier analysis to preserve only relevant information from previous time samples, reducing the effect that noisy signals can have on time series forecasting. In [199], it shows how a time series forecasting model fully based on MLP can compare competitively with state-of-the-art Transformer models (e.g., PatchTST [191]), reducing, in this way, the forecasting computational cost. In detail, it proposes an adaption (*TSMixer*) of *MLP-Mixer*, originally proposed for the vision domain, for time series forecasting. A convolution-based architecture, *MICN* (*Multi-scale Isometric Convolution Network*) [200], can discover local patterns and global correlations in time series by using a *multi-branch structure* and relying on *downsampled 1D convolutions* to extract local features and *isometric convolutions*, a particular case of causal convolution (see Section 3.1.2), to discover global correlations. Table 14 summarises the aforementioned models.

**Table 14.** Recent applications on time series forecasting using other deep learning architectures.

| Ref. | Year | Application |
|:---:|:---:|:---:|
| [197] | 2022 | Climate data/Electronic Health Records |
| [198] | 2022 | Long-term benchmark data sets (see Section 6.2) |
| [199,200] | 2023 | Long-term benchmark data sets (see Section 6.2) |

## 6. Benchmarks for Time Series Forecasting

Recently, a group of time series have emerged as benchmarks for assessing the performance of machine learning models in time series forecasting tasks. This section describes the most relevant benchmarks for both short and long-term forecasting.

### 6.1. Benchmarks for Short-Term Forecasting

Among several different data sets used for short-term forecasting, the most popular ones are described in Table 15. It is worth quoting the *M4* data set [44], proposed in 2020 for the homonymous *M4 competition* as a common benchmark for evaluating the performance of short-term forecasting models. The M4 data set contains 100.000 time series subdivided according to their data frequency into six groups: *M4-Yearly*, *M4-Quarterly*, *M4-Monthly*, *M4-Weekly*, *M4-Daily* and *M4-Hourly*. Furthermore, time series are also categorised into six domains, namely, *Demographic*, *Finance*, *Industry*, *Macro*, *Micro* and *Other*. Some insights on how time series are distributed into these categories are given in Figure 12.

**Table 15.** Short-term forecasting data sets. The column *Dim* refers to the dimensionality of time series.

| Dataset | Dim | Data Type (Real/Synthetic) |
|:---:|:---:|:---:|
| M4-Yearly [44] | 1 | Real |
| M4-Quarterly [44] | 1 | Real |
| M4-Monthly [44] | 1 | Real |
| M4-Weekly [44] | 1 | Real |
| M4-Daily [44] | 1 | Real |
| M4-Hourly [44] | 1 | Real |
| Mackey-Glass [201] | 1 | Synthetic |
| DatasetA [202] | 1 | Real |
| DSVC1 [203] | 1 | Real |
| Paris-14E [204] | 1 | Real |
| DatasetD [205] | 1 | Synthetic |

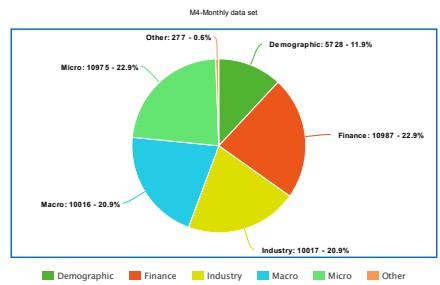

(**a**) M4-Monthly data set composition.

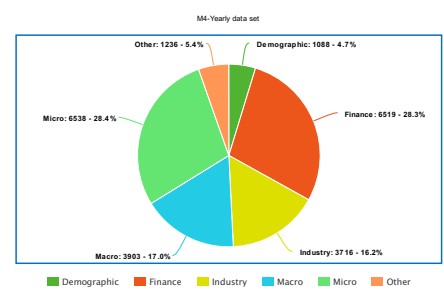

(**b**) M4-Yearly data set composition.

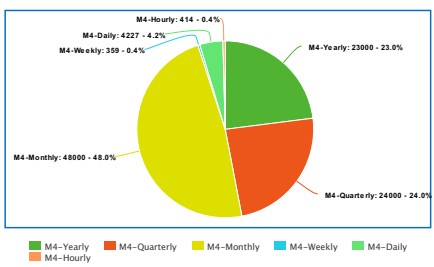

(**c**) M4-Competition data set composition.

**Figure 12.** Composition of the M4 data set. (**a**,**b**) show, respectively, the distribution of M4-Monthly and M4-Yearly time series into the six M4 categories. (**c**) shows the number of time series in each of the M4 data sets.

*6.2. Benchmarks for Long-Term Forecasting*

Nowadays, a group of specific data sets has become the *de-facto* benchmark [183] to assess long-term forecasting accuracy of all Transformer variants presented in Section 4.1.4. In detail, this benchmark is composed of nine multivariate time series data sets concerning the following domains: *electricity*, *transportation*, *weather*, *exchange rate* and *illness* (see Table 16). Time resolution can vary from 10 min up to 7 days.

**Table 16.** Long-term forecasting benchmark data sets. The data set size $(\cdot, \cdot, \cdot)$ refers to the cardinality of training, validation and test set, respectively. The columns *dim*, *pred len* and *time res* refer to the dimensionality of time series, the number of predicted future samples and the time resolution, respectively.

| Dataset | Dim | Pred Len | Dataset Size | Time Res | Domain |
|---|---|---|---|---|---|
| ETTm1 | 7 | [96,192,336,720] | (34,465, 11,521, 11,521) | 15 mins | Electricity |
| ETTm2 | 7 | [96,192,336,720] | (34,465, 11,521, 11,521) | 15 mins | Electricity |
| ETTh1 | 7 | [96,192,336,720] | (8545, 2881, 2881) | 15 mins | Electricity |
| ETTh2 | 7 | [96,192,336,720] | (8545, 2881, 2881) | 15 mins | Electricity |
| Electricity | 321 | [96,192,336,720] | (18,317, 2633, 5261) | 1 h | Electricity |
| Traffic | 862 | [96,192,336,720] | (12,185, 1757, 3509) | 1 h | Transport |
| Weather | 21 | [96,192,336,720] | (36,792, 5271, 10,540) | 10 mins | Weather |
| Exchange | 8 | [96,192,336,720] | (5120, 665, 1422) | 1 day | Finance |
| ILI | 7 | [24,36,48,60] | (617, 74, 170) | 1 week | Illness |

**7. Conclusions**

The paper has reviewed deep learning architectures for time series forecasting, underlining their advances. Nevertheless, four major problems remain open. The first one resides in the inability of most deep learning methods, with the exception of Deep Gaussian Processes, to estimate a confidence interval for the time series prediction. In principle, all deep learning architectures quoted in the survey can be properly modified using *Bayesian training strategies* [206] in order to provide the uncertainty of the model prediction, but, to the best of our knowledge, it has not been performed yet. The second problem resides in

the development of more and more complex deep learning architectures. This makes them subject to overfitting, a problem that can hardly be faced by deep learning architectures. Therefore, the development of deep learning architectures for time series forecasting that are robust w.r.t. overfitting is becoming more and more relevant. The third problem consists in the need for adequately long time series. In particular, some deep learning architectures, e.g., Transformers, require the estimation of millions of parameters, implying, in this way, the necessity of adequately long time series for estimating them. The problem seems to be partially addressed by data augmentation but the proposed solutions are not fully adequate, yet. Finally, the last problem emerges in most of the reviewed deep learning models. They assume the dynamical stationarity of time series, implying that the dynamic system generating time series is stationary over time. When the aforementioned assumption is violated, a *concept drift* phenomenon [207] in time series is observed, consequently leading to a dramatic decrease in the prediction accuracy of deep learning models for time series forecasting.

**Author Contributions:** Conceptualization, A.C., V.C. and F.C.; methodology, A.C. and V.C.; validation, A.C., V.C. and F.C.; formal analysis, F.C.; investigation, A.C., V.C. and G.I.; resources, A.C., V.C., G.I. and F.C.; writing—original draft preparation, A.C., V.C. and F.C.; writing—review and editing, A.C., V.C. and F.C.; visualization, A.C., V.C. and F.C.; supervision, F.C.; project administration, F.C. All authors have read and agreed to the published version of the manuscript.

**Funding:** This research received no external funding.

**Data Availability Statement:** No new data were created or analyzed in this study. Data sharing is not applicable to this article.

**Acknowledgments:** We would like to thank the anonymous reviewers for the useful comments.

**Conflicts of Interest:** The authors declare no conflict of interest.

## Appendix A. Table of Mathematical Expressions

In Appendix A, the table of the most frequent mathematical expressions and operations is provided. Table A1 provides a description for each mathematical operation.

**Table A1.** Table of the most commonly used mathematical operations with their respective description.

| Symbol | Definition |
| --- | --- |
| $\mathcal{Y} = w * \mathcal{X}$ | Convolution between a kernel $w$ and a sequence $\mathcal{X}$. The result is a new sequence $\mathcal{Y}$. |
| $\vec{y} = \vec{u} \odot \vec{v}$ | Element-wise product between two vectors $\vec{u}$ and $\vec{v}$. The result is a vector $\vec{y}$ such that $y_i = u_i v_i$. |
| $V \otimes W$ | Tensor product between two vectors $V$ and $W$, the result is a matrix. |
| $\mathbb{I}$ | The Identity matrix. |

## Appendix B. Diffusion Models

In this section, the most relevant diffusion models, i.e., DDPMs (Appendix B.1), SGMs (Appendix B.2) and SDEs (Appendix B.3), and foundations, are described.

*Appendix B.1. Denoising Diffusion Probabilistic Models*

*Denoising Diffusion Probabilistic Models*, originally proposed in [173] and later extended in [174], consider two Markov chains for forward and backward process, respectively, to approximate the generative process of observed data. In detail, let the original noiseless data be $x^0$. The forward Markov chain projects $x^0$ into a sequence of noised data $x^1, x^2, \dots, x^K$ with a *diffusion transition kernel*:

$$q(x^k|x^{k-1}) = \mathcal{N}(\sqrt{a_k}x^{k-1}, (1 - a_k)\mathbb{I}), \tag{A1}$$

where $K$ is the finite number of noise level of forward process, $a_k \in (0,1)$ for $k = 1,2,\ldots,K$ are hyperparameters indicating the variance of the noise level at each step, $\mathcal{N}(\cdot,\cdot)$ is the Gaussian distribution, and $\mathbb{I}$ is the identity matrix. The Gaussian transition kernel $q(\cdot|\cdot)$ has a fundamental property that allows obtaining $x^k$ directly from original sample $x^0$:

$$q(x^k|x^0) = \mathcal{N}(\sqrt{\tilde{a}_k}x^0, (1 - \tilde{a}_k)\mathbb{I}), \tag{A2}$$

where $\tilde{a}_k = \prod_{i=1}^{k} a_i$. In DDPM, the *reverse transition kernel* $p_\theta(\cdot|\cdot)$ is designed by a deep neural network:

$$p_\theta(x^{k-1}|x^k) = \mathcal{N}(\mu_\theta(x^k,k), \Sigma_\theta(x^k,k)), \tag{A3}$$

where $\theta$ indicates learnable parameters of deep neural networks. In order to compute the parameters $\theta$ such that samples estimated by $p_\theta(x^0)$ are almost identical to observed data $x^0$, a *maximum likelihood estimation* method is performed, minimising the variational lower bound of the estimated negative log-likelihood $\mathbb{E}[-\log p_\theta(x^0)]$:

$$\mathbb{E}[-\log p_\theta(x^0)] = \mathbb{E}_{q(x^{0:K})}\left[-\log p(x^K) - \sum_{k=1}^{K} \log \frac{p_\theta(x^{k-1}|x^k)}{q(x^k|x^{k-1})}\right], \tag{A4}$$

where $x^{0:K}$ indicates the sequence $x^0,\ldots,x^K$. A simpler objective function [174] can be provided, as follows:

$$\mathbb{E}_{k,x^0,\epsilon}[\delta(k)||\epsilon - \epsilon_\theta(\sqrt{\tilde{a}_k}x^0 + \sqrt{1 - \tilde{a}_k}\epsilon, k)||^2], \tag{A5}$$

assuming the covariance matrix equal to $\Sigma_\theta(x^k,k) = \sigma_k^2\mathbb{I}$, where $\sigma_k^2$ controls the noise level and may vary at different reverse steps, and $\delta(k) = \frac{(1-a_k)^2}{2\sigma_k^2 a_k(1-\tilde{a}_k)}$.

*Appendix B.2. Score-Based Generative Models*

　　*Score-based generative models* (SGMs) [175], are made up of two modules. The former is the *score matching module* [208], for estimating the unknown target distribution $q(x)$ with the *Stein score* approximation, $\nabla_x \log q(x)$, by means of a *score-matching network* (e.g., *denoising score matching* [209], *slided score matching* [210]). The latter is the *annealed Langevin dynamics* (ALD), that is a sampling algorithm generating samples with an iterative *Langevin Monte Carlo* process at each update step. The fundamental idea behind *denoising score matching* is to process the observed data $x^0$ with the *forward transition kernel* $q(x^k|x^0) = \mathcal{N}(x^0, \sigma_k^2\mathbb{I})$, and to estimate jointly the Stein scores for the *noise density distributions* $q_{\sigma_1}(x), q_{\sigma_2}(x),\ldots,q_{\sigma_k}(x)$. In this case, the Stein score for *noise density function* $q_{\sigma_k}(x)$ is $\nabla_x \log q_{\sigma_k}(x)$. Hence, a neural network $s_\theta(x,\sigma_x)$, with learnable parameters $\theta$, can approximate the Stein score. The initial objective function is therefore given by:

$$\mathbb{E}_{k,x^0,x^k}[\delta(k)||s_\theta(x^k,k) - \nabla_x^k \log q_{\sigma_k}(x^k)||^2]. \tag{A6}$$

Subsequently, the ALD algorithm is used for the sampling phase. The algorithm is initialised with a sequence of increasing noise levels $\sigma_1,\ldots,\sigma_K$ and a starting sample $x^{K,0} \sim \mathcal{N}(0,\mathbb{I})$. For $k = K, K-1,\ldots,0$, $x^k$ is updated with $N$ iterations that compute:

$$z \leftarrow \mathcal{N}(0,\mathbb{I}) \tag{A7}$$

$$x^{k,n} \leftarrow x^{k,n-1} + \frac{1}{2}\psi_k s_\theta(x^{k,n-1},\sigma_k) + \sqrt{\psi_k}z, \tag{A8}$$

where $n = 1,\ldots,N$ and $\psi_k$ represent the step of update.

*Appendix B.3. Stochastic Differential Equations*

As described in Appendices B.1 and B.2, both DDPMs and SGMs models define a *discrete forward process*, considering *N* iterations of diffusion steps. In order to define a *continuous diffusion process*, a solution based on *stochastic differential equations* (SDEs) [176] has been proposed. Since both DDPMs and SGMs are discrete forms of SDEs, the SDEs formulation can be considered as an extension of the aforementioned two definitions. Therefore, the *backward process* is modelled as a *time-reverse SDE* (see Equation (A10)), and the samples can be generated by solving time-reverse SDE. A general expression of SDE is defined as follows:

$$dx = f(x,k)dk + g(k)dw, \tag{A9}$$

and the time-reverse SDE [211] is:

$$dx = [f(x,k) - g(k)^2 \nabla_x \log q_k(x)]dk + g(k)d\tilde{w}, \tag{A10}$$

where $w$ and $\tilde{w}$ are *standard Wiener processes* [212]. It can be proved [176] that the sampling from the *probability flow ordinary differential equations* (ODE) yields the same distribution of the time-reverse SDE:

$$dx = [f(x,k) - \frac{1}{2}g(x)\nabla_x \log q_k(x)]dk, \tag{A11}$$

where $f(x,k)$ and $g(k)$ are the *drift* and *diffusion* coefficients for the diffusion process, respectively, and $\nabla_x \log q_k(x)$ is the Stein score that can be learned with similar methods as in SGMs (see Appendix B.2). At this point, it can observed that the DDPMs can be reformulated in terms of SDEs, that generally known as *variance preserving* (VP) SDE [176]. The same reformulation can be done for the forward process of SGMs, where the corresponding SDE is known as *variance exploding* (VE) SDE [176]. After having learned the score model $s_\theta(x,k)$, the samples are generated by solving the time-reverse SDE or the probability flow ODE with ALD techniques.

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
