# Peer review of "Deep Learning for Time Series Forecasting: Advances and Open Problems"

_information, doi:10.3390/info14110598_

Round 1
Reviewer 1 Report
Comments and Suggestions for Authors
In this study, deep learning models for time series forecasting are introduced, and their features are summarized. However, I think certain changes are required:
1.It is required to quickly describe the current approaches for using deep learning with time series and each one's unique properties in the Introduction section.
2. Detailed definitions and descriptions of short-term forecasting must be given in Section 3, "Deep Learning Models for Short-term Forecasting," along with a list of existing models.
3. Similarly, detailed definitions and descriptions of long-term forecasting are required in Section 4, "Deep Learning Models for Long-term Forecasting," along with a list of existing models.
4. Using adverbs like "slightly" that express subjective feelings in Section 4.1.2, "Multi-Head Attention," may make the article less accurate. To help readers understand why and how MHA is more advanced, the structural diagram of MHA can be included in it and contrasted to the scaled dot product attention.
5. Additional transformer variants from diverse domains can be added to Section 4.1.3, "Transformer Variants for Time Series Forecasting," such as Earthformer, which focuses on predicting the Earth's system. Since many traditional Transformer methods fall short in predicting river flows and floods, this research offers a lot of potential.
Reviewer 2 Report
Comments and Suggestions for Authors
1. Abstract:
- Incorporate Key Findings: Expand the abstract to include 1-2 primary conclusions or major insights, ensuring readers get the crux of the paper at a glance.
2. Introduction:
- Value Statement: Explain how this review addresses gaps missed by previous studies.
- Sentence Refinement: Replace "applying deep learning to time series forecasting is straightforward" with a more nuanced statement reflecting challenges and advancements.
3. Deterministic Time Series:
- Terminology Clarification: When introducing terms like 'univariate' or 'multivariate,' provide short, parenthetical definitions.
- Visualization: Incorporate a graphical representation of deterministic time series and the autoregressive model.
4. Deep learning models for short-term forecasting:
- Real-world Implications: Choose 1-2 applications and delve deeper into the real-world impacts of using the mentioned method for those applications.
- Comparative Analysis: Feature a table or chart comparing methods based on accuracy, efficiency, and scalability.
- Challenges and Limitations: Dedicate a subsection to discuss common hurdles like overfitting, data requirements, and interpretability.
5. Deep learning models for long-term forecasting:
- Terminology Assistance: Introduce complex terms with brief explanations or footnotes.
- Illustrative Examples: Add case studies highlighting where specific Transformer variants showed superior results.
- Visual Aid: Insert flowcharts or diagrams illustrating the various Transformer architectures.
- Performance Metrics: Showcase a table detailing the performance (e.g., RMSE, MAPE) of Transformer variants on standard datasets.
6. Other relevant deep learning models:
- Provide More Depth: Elaborate on models' distinctive features and relevancy, like the MLP-Mixer in time series forecasting.
7. Benchmarks for time series forecasting:
- Graphical Representation: Introduce bar graphs or pie charts detailing the data distribution of datasets like M4-Yearly.
- Dataset Popularity: Offer a short note on the factors leading to the popularity of datasets, especially the M4 series.
8. Conclusions:
- Solutions to Challenges: For each identified challenge, propose or hint at future research directions or explanations.
- Emphasize Practical Impacts: Elucidate on real-world consequences of challenges, such as the requirement for extensive time series data.
The paper is generally well-written, with a clear structure and logical flow. However, there are several instances where phrasing could be improved for clarity. Additionally, some sentences are overly lengthy, which can distract from the core message. It would benefit from thorough proofreading to ensure that the intended meaning is consistently clear and the language is concise.
Reviewer 3 Report
Comments and Suggestions for Authors
In this paper, authors a review of state-of-the-art deep learning architectures for time series forecasting, underlining recent advances and open problems, and also paying attention to benchmark data sets. The discussed issue shows the research significance, and the structure is relative good. However, minor revisions are needed before the acceptance.
1. It is proven the current Transformer neural networks can achieve the best performance for time series data, so it is suggested to include the performance discussion on Transformer in this paper.
2. How to deploy the testing platform to evaluate different learning algorithms, the discussion can be added.
3. For the evaluation, the dataset/algorithm choosing reasons, the detailed platform configurations and the discussion on other untested datasets should be introduced in the revised paper.
4. It is better to provide a notation table covering frequently used math symbols.
5. Please go through the paper carefully and double check whether the right template are used. Correct some typos and formatting issues (e.g., “3. Deep learning models for short-term forecasting” -> “3. Deep Learning Models for Short-Term Forecasting”?).
6. Some references lack the necessary information (e.g., [25]?), please provide all information according to the right template.
7. Make the References more comprehensive, besides this work, some other promising scenarios (e.g., Big data, other IoT systems) can be covered in this work. If the above related work can be discussed, it can strongly improve the research significance. For the improvement, the following papers can be considered to make the references more comprehensive.
J. Wang, Y. Zou, P. Lei, et al. Research on Recurrent Neural Network Based Crack Opening Prediction of Concrete Dam. Journal of Internet Technology, 2020, 21(4):1151-1160
J. Wang, Y. Yang, T. Wang, R. Sherratt, J. Zhang. Big Data Service Architecture: A Survey. Journal of Internet Technology, 2020, 21(2): 393-405
J. Zhang, S. Zhong, T. Wang, H.-C. Chao, J. Wang. Blockchain-Based Systems and Applications: A Survey. Journal of Internet Technology, 2020, 21(1): 1-14
W. Wang, Y. Yang, J. Li, Y. Hu, Y. Luo, X. Wang: Woodland Labeling in Chenzhou, China, via Deep Learning Approach. Int. J. Comput. Intell. Syst. 13(1): 1393-1403 (2020)
Comments on the Quality of English LanguageThe English writing quality is good.
Reviewer 4 Report
Comments and Suggestions for Authors
The article lists and classifies deep learning models through long-term time series prediction and short-term time series prediction. It also explains the time series characteristics of each model, the principle and application scope of the model. The article is overall complete, logically clear, and discusses current hot topics. But there are a few more points that need to be added:
1. The review is not comprehensive enough, and some typical deep learning models are not listed. Generative Adversarial Networks can also be used for time series prediction.
2. Review should not only list the corresponding literature, but also conduct in-depth research, actively digest these literatures, and summarize the shortcomings of the model that you understand rather than directly copying.
3. In the conclusion section at the end of the article, after summarizing the problems, it is recommended to add a section on development trends or possible solutions. This can reflect the author's understanding of the entire field.
4. Do a literature search again and add a few more recent articles.
5. Format issues, such as the indentation of the first line of a paragraph, need to be rechecked.
6. In the model structure figures, the input variable fonts are unified into italics.
7. Add appropriate transition sentences between each model introduction part. Make the logic clearer.
8. It is better to list the shortcomings of each model.
Comments on the Quality of English LanguageGood
Reviewer 5 Report
Comments and Suggestions for Authors
In this paper the authors compare deep learning architectures for time series forecasting, underlining recent advances and open problems, and also paying attention to benchmark data sets. Some points should be included within the manuscript in order to improve the publication.
- The aim of the authors is a review paper and therefore they analyze a lot of references.
- The abstract should be also rewritten in a more successive way. Some comments about the results of this work are necessary.
- More details about the problem and the aim of this work should be written in the introduction section.
- In the conclusions section, the authors should describe the contribution of their study in more detail.
- The authors must include some related references, such as the following:
- Marta Narigina, Arturs Kempelis, Andrejs Romanovs, "Machine Learning-based Forecasting of Sensor Data for Enhanced Environmental Sensing," WSEAS Transactions on Systems, vol. 22, pp. 543-555, 2023.
- Amin Karimi Dastgerdi, Paolo Mercorelli, "Investigating the Effect of Noise Elimination on LSTM Models for Financial Markets Prediction Using Kalman Filter and Wavelet Transform," WSEAS Transactions on Business and Economics, vol. 19, pp. 432-441, 2022.
Round 2
Reviewer 2 Report
Comments and Suggestions for Authors
Reviewing the revised manuscript, I can confirm that the authors have improved. As with many evaluations, time provides clarity and perspective. Looking back, it's evident that the paper has evolved greatly and now stands as an excellent piece of research. Researchers must be meticulous in their approach; this work is no exception. The dedication and effort invested in this research are admirable. As time passes, challenges become memories, and this paper has certainly transformed for the better.
Reviewer 4 Report
Comments and Suggestions for Authors
The authors have dealt with all my concerns.
Reviewer 5 Report
Comments and Suggestions for Authors
The paper can be published in its present form.